

# Combining probability distributions of sea level variations and wave run-up to evaluate coastal flooding risks

Ulpu Leijala, Jan-Victor Björkqvist, Milla M. Johansson, Havu Pellikka, Lauri Laakso, and Kimmo K. Kahma

Finnish Meteorological Institute, P.O. Box 503, FI-00101, Helsinki, Finland

*Correspondence to:* Ulpu Leijala (ulpu.leijala@fmi.fi)

**Abstract.** Tools for estimating probabilities of flooding hazards caused by the simultaneous effect of sea level and waves are needed for the secure planning of densely populated coastal areas that are strongly vulnerable to climate change. In this paper we present a method for combining location-specific probability distributions of three different components: 1) long-term mean sea level change, 2) short-term sea level variations, and 3) wind-generated waves. We apply the method in two locations in the Helsinki Archipelago to obtain run-up level estimates representing the joint effect of the still water level and the wave run-up. These estimates for the present, 2050 and 2100 are based on field measurements and mean sea level scenarios. In the case of our study locations, the significant locational variability of the wave conditions leads to a difference in the safe building levels of up to one meter. The rising mean sea level in the Gulf of Finland and the uncertainty related to the associated scenarios contribute significantly to the run-up levels for the year 2100. We also present a sensitivity test of the method and discuss its applicability to other coastal regions. Our approach allows for the determining of different building levels based on the acceptable risks for various infrastructure, thus reducing building costs while maintaining necessary safety margins.

## 1 Introduction

Predicting coastal flooding and extreme sea level events has a focal role in designing of rapidly evolving coastal areas, that become continuously more populated and convoluted. Such flooding events, related to extreme sea levels, are influenced by long-term changes in mean sea level, together with short-term sea level variations and the wind-generated wave fields. These processes are further influenced by a variety of other processes and conditions like the vertical crustal movements, islands, the shape of the shoreline and the topography of the seabed. Because of a rising mean sea level, the effect of sea level variations accompanied by waves might cause more damage in the future.

**Globally**, several studies have addressed the topic of combining sea level changes and variations with wind waves in different circumstances and locations, using different methods and assumptions. Hawkes et al. (2002) studied the combined effect of large waves and high still water in coastal areas of England and Wales using Monte Carlo simulations, accounting for the dependence between the water level, the wave height and the wave steepness. Hawkes (2008) summarized joint probability methods and discussed issues related to data selection and event definition, concluding that the analysis method and source data should be well chosen to meet the requirements of a particular problem.



Wahl et al. (2012) applied Archimedean Copula functions in the German Bight to achieve exceedance probabilities for storm surges and wind waves. They found that using this methodology, realistic exceedance probabilities can be achieved and used to enhance the results from integrated (i.e. multivariate problems) flood risk analyses. A copula based approach was also implemented by Masina et al. (2015) to examine the joint distribution of sea level and waves at a location suffering from coastal

flooding in the northern Italy (Ravenna coast). This method accounts for the dependence structure between the variables, and the authors also assessed the present probability of marine inundation accounting for the interrelationship among the main sea condition variables and their seasonal variability. Results of this study highlight the need to utilize all variables and their dependences simultaneously for obtaining realistic estimates for flooding probabilities.

In a study conducted by Prime et al. (2016), the authors used a combination of a storm impact model and a flood inundation

model to quantify the uncertainty in flood depth and extent of a 0.5% probability event in the Dungeness and Romney Marsh coastal zone in the UK. They found that the most significant flood hazards on their study site were caused by low swell waves during highest water levels, as opposed to large wind waves occurring at lower water levels. Chini and Stansby (2012) used an integrated modelling system to investigate the joint probability of extreme wave height and water level at Walcott on the eastern coast of the UK, thus determining changes in overtopping rates. Using different scenarios for the mean sea level rise,

the authors found that flooding probabilities are mainly influenced by changes in water level, as opposed to changes in the waves conditions. Cannaby et al. (2016) reached a similar conclusion when studying coastal flooding risks in the Singapore region.

Although the changes in water level have been deemed to have to highest impact on flooding risks by several authors, Chini et al. (2010) found the near-shore wave conditions in the East Anglia coast (UK) to be sensitive to the changes in water level.

The authors used five linear sea level rise scenarios, and one climatic scenario for storm surges and offshore waves to study the waves between 1960 and 2099. Cheon and Suh (2016) also found that the depth-limitation of waves can be relaxed with increasing mean sea level, thus leading to increased risks for wave-induced damages on inclined coastal structures.

**The Baltic Sea** is a shallow semi-enclosed marginal sea, connected to the Atlantic Ocean only through the narrow and shallow Danish Straits. This gives the sea level variations in the Baltic Sea an unique nature, which differs from that on the

ocean coasts. The components of local sea level variations in a short time scale include wind waves, wind and air pressure induced sea level variations, currents, tides, internal oscillation (seiche) and meteotsunamis. Long-term changes are related to the climate change driven mean sea level variations, postglacial land-uplift, and the limited exchange of water through the Danish straits, which causes variations up to 1.3 m in the average level of the Baltic Sea on a weekly time scale (Leppäranta and Myrberg, 2009; Pellikka et al., 2014; Johansson et al., 2014).

Both sea level and wind waves have been studied thoroughly separately in the Baltic Sea area, but research into their joint effect is sparse compared to coastal regions outside of the Baltic Sea. Hanson and Larson (2008) examined jointly waves and water levels to estimate run-up levels (as the sum of the mean water level and the wave run-up height) on the Swedish coast in the southern Baltic Sea. They established probability distributions based on existing climate data (mainly wind and water level data) including also scenarios of future climate change. The impact of breaking waves on the mean water level (wave set-up)

has been studied on the Estonian coast in the Gulf of Finland. Based on results from a numerical wave model, Soomere et al.





(2013) found the wave set-up to be strongly affected by the wind direction. Pindsoo and Soomere (2015) reached the same conclusion in a study that also accounted for varying offshore water level variations simulated by the Rossby Centre Ocean (RCO) model.

**In Finland**, there is a clear demand for flooding risk evaluation. The irregular coastline of approx. 46 000 km is characterised
by coastal archipelagos consisting of about 73 000 islands. Especially the southern part of the coast might become exposed to increasing flooding risks, since its rather small land-uplift rate will no longer cancel out the accelerating mean sea level rise (Pellikka et al., 2017).

The previous sea level records at almost all of the tide gauges along the Gulf of Finland were exceeded in 2005 during the storm Gudrun. In that flooding event, three different components acted simultaneously: a high total water amount in the Baltic
Sea, a high phase of the standing waves (seiches), and severe winds piling up the water and waves towards the shore. Gudrun caused major damage to coastal infrastructure on both north and south sides of the Gulf of Finland (Parjanne and Huokuna, 2014; Tõnisson et al., 2008; Suursaar et al., 2006). The coastal floods, especially in the Gulf of Finland, seem to be the most severe hazard among the extreme sea events (storm surges etc.) in the Baltic region (Kulikov and Medvedev, 2013).

The earlier flooding risk estimates in Finland (Kahma et al., 1998, 2014; Pellikka et al., 2017) were based on combining
the probability distributions of the observed short-term sea level variability and the long-term mean sea level projections (Johansson et al., 2014). On top of these estimates for the sea level variations up to 2100, a location-specific additional height for wind waves (henceforth "wave action height") was accounted for separately.

In this study, we use a probabilistic method to calculate the joint effect of the above mentioned two components of sea level, and wind waves. We utilize location-specific probability distributions of water level and wave run-up (the maximum
vertical elevation of the water in relation to the still water level during a certain period). We aim at getting a single probability distribution for the maximum absolute elevation of the continuous water mass (Fig. 1). For simplicity, we call this resulting elevation the run-up level. The method presented in this paper has been applied to assess the safe building heights on the coast of Helsinki (Kahma et al., 2016).

This paper is structured in the following manner. In Sect. 2, we outline the parameters affecting the sea surface level on
the Finnish coast. In Sect. 3, we introduce the scenarios and observations used in this study. This is continued in Sect. 4 by forming the sea level and wave probability distributions, presenting the theory for evaluating the sum of two random variables, and the particulars of applying it to sea level variations and wind waves. We then investigate the sensitivity of the approach on the properties of the probability distributions by applying it to different theoretical wave distributions with known parameters in Sect. 5. We apply the method on a case study in the Helsinki Archipelago in Sect. 6. The paper is finished by discussion on
the relevance and applicability of the results in Sect. 7, and finally conclusions in Sect. 8.




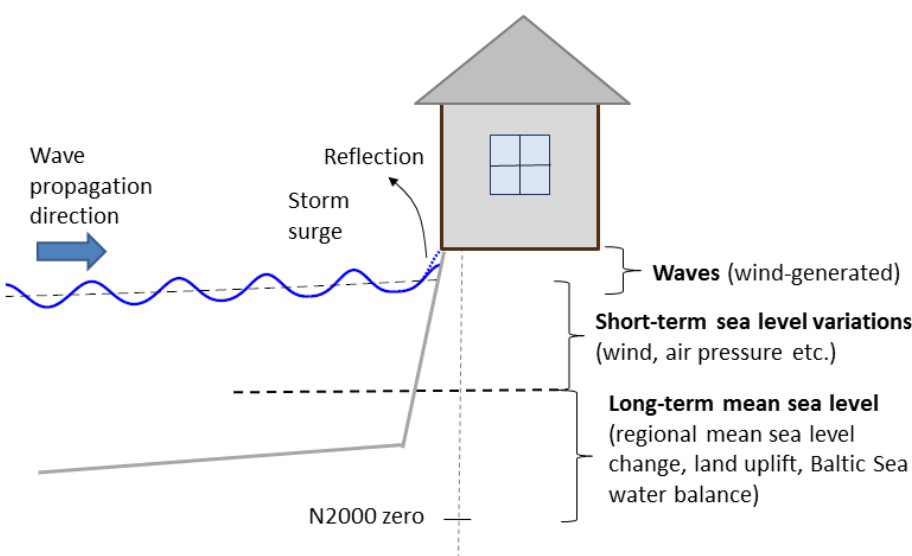

**Figure 1.** The run-up level i.e. the maximum absolute elevation of the continuous water mass (solid blue) is a result of the 1) long-term mean sea level change, 2) short-term sea level variations and 3) wind-generated waves. On a steep shore the waves can also be fully or partially reflected (dotted blue).

## 2   Components contributing to the sea surface level

The instantaneous sea surface height at any coastal site in the Baltic Sea is affected by several physical processes in different time scales. In this study, we present the local sea surface height or, more precisely, the elevation to which the continuous water mass reaches, $H$, as a sum of three components with different time scales (Fig. 1):

$$H = S_L + S_S + H_{runup} \tag{1}$$

where $S_L$ stands for the long-term sea level, $S_S$ for the short-term sea level, and $H_{runup}$ for the wave height above the still water level ($S_L + S_S$). For clarity, we use the order: 1) long-term sea level, 2) short-term sea level, 3) waves throughout the paper.



**The long-term mean sea level** changes slowly from decade to decade. On the Finnish coast it is affected by the global mean sea level, the post-glacial land uplift and the Baltic Sea water balance (Johansson et al., 2014). The global mean sea level is rising due to thermal expansion and the melting of e.g. the Antarctic and the Greenland ice sheets. Nevertheless, the rising sea level is locally mitigated by the post-glacial land uplift, which presently amounts to 3–10 mm/yr on the Finnish coast. The

mean sea level in the Baltic Sea can also deviate from the mean ocean level because of the limited water exchange through the narrow and shallow Danish Straits, which connect the Baltic Sea to the North-Atlantic Ocean. The Baltic Sea water balance is mainly controlled by the wind and air pressure conditions over the Baltic Sea and North Sea areas.

**Short-term water level variations** on sub-decadal time scale on the Finnish coast range from -1.3 m to +2.0 m above the long-term mean sea level, with time scales ranging from year-to-year variability of the Baltic Sea total water volume down to

storm surges and other rapid variations in less than an hour. The week-to-week variability of the water volume results into a sea level variability of about 1.3 m, while the shorter-period internal variations in the Baltic Sea basin contribute several tens of centimetres to the sea level variability (Leppäranta and Myrberg, 2009). Along the Finnish coast, the largest variations occur near the closed ends of the Bay of Bothnia and Gulf of Finland, while the range of variability on sites closer to the central area of the Baltic Sea is substantially smaller (e.g. Johansson et al., 2001). These variations are mainly driven by wind and air

pressure variations. Ice conditions in the winter also affect the water level variability, but unlike in many other coastal areas, the tidal variations are only a few centimeters to 10–15 centimeters on the Finnish coast (Witting, 1911; Leppäranta and Myrberg, 2009; Särkkä et al., 2017).

**The wave conditions** in the Baltic Sea are influenced by the limited fetch, the topography of the seabed and the seasonal ice-cover (Tuomi et al., 2011). The highest observed significant wave height in the Baltic Sea is 8.2 m (Björkqvist et al., 2017b).

In the Gulf of Finland the growth of the waves is restricted by the narrowness of the gulf (Kahma and Pettersson, 1994), but a significant wave height of 5.2 m has still been measured in the centre of the Gulf of Finland (Pettersson et al., 2013). Close to the shoreline the waves are modified by the archipelago and the irregular shoreline (Tuomi et al., 2014; Björkqvist et al., 2017a). The significant wave height close to the coast in the Helsinki archipelago has been estimated to not exceed 2 m (Kahma et al., 2016), but the steep shoreline near Helsinki causes wave reflection leading to a positive interference (Björkqvist et al.,

2017c). This wave reflection affects the value of the wave run-up, which is the vertical elevation during a certain time (with respect to the still water level).

## 3 Scenarios and observations used in this study

To estimate the three components contributing to the coastal sea surface height (Eq. 1) in the present conditions and in the future, we used a combination of literature-based scenarios and observations.

### 3.1 Long-term mean sea level: past estimate and future scenarios

We focused our calculations on three different time instants: the present year 2017, and future years 2050 and 2100. Pellikka et al. (2017) calculated estimates for the past long-term mean sea level, as well as future scenarios, on the Finnish coast. They





**Figure 2.** The coastal area off Helsinki and the measurement sites used in the study. The red box in the Baltic Sea map (top) marks the area shown on the bottom. The circles mark the location of the moored wave buoys and the star represents the Helsinki tide gauge used to collect the sea level data. The contours mark the approximate water depth.




estimated the past and present long-term mean sea level as a combination of the past actualised global sea level rise, land uplift, and the Baltic Sea water balance. The significant year-to-year variability in the Baltic Sea water balance was smoothed out by a 15-year floating average.

The future scenarios of Pellikka et al. (2017) were based on an ensemble of 14 global mean sea level rise predictions from the recent scientific literature. Each prediction was adjusted to the Finnish coast by taking into account the uneven geographical distribution of the thermal expansion of sea water, ocean dynamical changes, and the fingerprints of the melting ice masses. The regionalized predictions, along with their uncertainties, were combined to obtain a probability distribution of future sea level rise in 2000–2100. Lastly, these localized sea level rise scenarios were combined with the postglacial land uplift and an estimate of wind-induced changes in the Baltic Sea water balance. For more details of the method, see Johansson et al. (2014) and Pellikka et al. (2017). In Helsinki, the change in mean sea level in 2000–2100 was predicted to be 30 cm (–15 cm ... 87 cm, 5–95% uncertainty range).

### 3.2 Observed short-term sea level variability

The Finnish Meteorological Institute (FMI) operates 14 tide gauges along the Finnish coast, most of which have been operating since the 1920s. We used sea level observations from the Helsinki tide gauge, which started operation in 1904. Hourly sea level observations from Helsinki are available in digital format since 1971, providing a continuous 46 year (1971–2016) data set of instantaneous hourly sea level values. The Finnish sea level data are measured in relation to a tide gauge specific fixed reference level, which is regularly levelled to the height system N2000. The height system N2000 is a Finnish realization of the common European height system. The N2000 datum is derived from the NAP (Normaal Amsterdams Peil, Saaranen et al., 2009). For a more detailed description of the tide gauge data, measurement techniques and quality, see Johansson et al. (2001).

The sea level variations are location-specific, but as our study area is limited to sites less than 5 km away from the Helsinki tide gauge, we considered the sea level variability measured at the tide gauge sufficiently representative for both study sites at Jätkäsaari and Länsikari (Fig. 2).

### 3.3 Wind wave data

FMI conducts operational wind wave measurements in four locations in the Baltic Sea. In the Gulf of Finland, the observations are carried out using a Datawell Directional Waverider moored in the centre of the gulf (see Fig. 2). However, these open sea observations are not representative of nearshore wave conditions (e.g. Kahma et al., 2016; Björkqvist et al., 2017a). The operational measurements have therefore been supported by short-term observations with smaller Datawell G4 wave buoys inside the Helsinki archipelago.

We used the open sea measurements from the operational Gulf of Finland wave buoy in 2000–2014 in combination with shorter time series at chosen locations inside the Helsinki coastal archipelago. The measurements in the archipelago were conducted at Jätkäsaari (31 days in October 2012) and Länsikari (11 days in November 2013) (see Fig. 2). These shorter measurements were a part of a research project commissioned by the City of Helsinki (Kahma et al., 2016).





We chose the measurement sites at Jätkäsaari and Länsikari so that they would represent two different kinds of wave conditions: Jätkäsaari is close to the shore, in a place well sheltered from the open sea by islands. Länsikari, on the other hand, is located in the outer archipelago, relatively unsheltered from the open sea conditions.

We determined the significant wave height $H_s$ from the 26 minute wave buoy time series as $H_s = 4\sqrt{\sigma^2}$, where $\sigma^2$ is the variance of the wave buoy's vertical displacement. The significant wave height is a statistical parameter representing the height of the waves during a certain time or area, and it approximately corresponds to the wave height estimated by an experienced mariner.

## 4 Probability methods to combine sea level variations and wind waves

As a first step in estimating the combined effect of the long-term mean sea level, the short-term sea level variability, and the wind waves on the frequencies of exceedance of coastal floods, we constructed probability distributions for each of them separately (Sect. 4.1–4.3). Next, we calculated the probability distributions of their sum: the method for this is presented in Sect. 4.4, and applied on the three constructed distributions in Sect. 4.5.

In this paper, we use three types of probability distributions. The probability density function (pdf) $f_x$, the cumulative distribution function (cdf) $F_x$, and the complementary cumulative distribution function (ccdf) $\overline{F}_x$ of the random variable $\underline{x}$ are defined as:

$$
\begin{aligned}
f_x(x) &= P(\underline{x} = x) \\
F_x(x) &= P(\underline{x} \leq x) \\
\overline{F}_x(x) &= P(\underline{x} > x) = 1 - F_x(x)
\end{aligned}
\tag{2}
$$

Since our data is based on hourly values, we calculated the frequencies of exceedance from the ccdf by multiplying the probabilities by the average number of hours per year (8766). By using hourly sea level values we practically assume a constant sea level for the entire hour. When summing a one hour constant sea level value with a one hour maximum wave run-up elevation with respect to the mean water level, the result is the maximum absolute elevation within one hour. This maximum absolute elevation during one hour is defined as one event.

We use the term *still water level* to represent the maximum elevation of the water level (incl. short- and long-term sea level variations) corresponding to a certain frequency of exceedance. Moreover, we use the term *run-up level* to represent the maximum elevation of continuous water mass caused by the joint effect of sea level and waves corresponding to a certain frequency of exceedance.

### 4.1 Distributions of the long-term sea level scenario

The probability distributions for the long-term mean sea level scenarios on the Finnish coast were calculated by Pellikka et al. (2017). We used their pdfs for sea level scenarios at Helsinki in 2050 and 2100 (Fig. 3). The medians of these scenarios predict





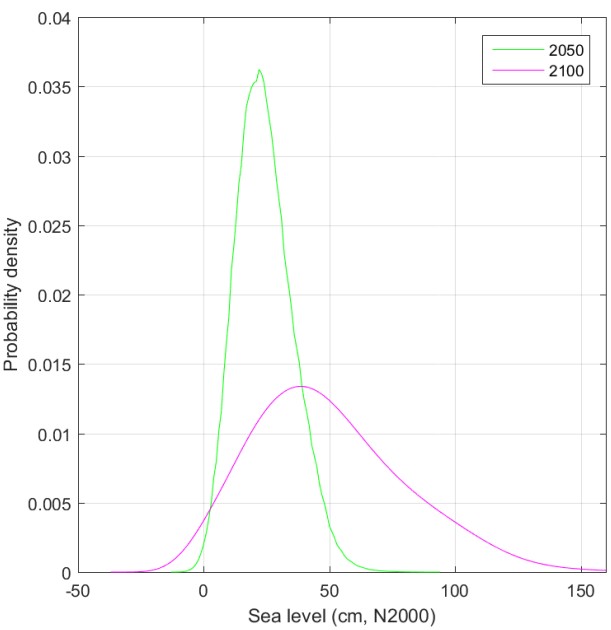

**Figure 3.** Probability density functions of future mean sea level at the Helsinki tide gauge for years 2050 and 2100; from Pellikka et al. (2017).

a rise of 4 cm from the estimated mean sea level of 2017 (+19 cm) up to 2050, and a rise of 28 cm from 2017 to 2100. The uncertainties, however, increase markedly in the future, the width of the 5% to 95% range of the cdf being 37 cm in 2050 and 103 cm in 2100.

## 4.2 Distributions of the short-term sea level variability

5 We constructed the probability distribution of short-term sea level variability from the observed sea levels in 1971-2016. The observed sea levels, from which the wave action has been filtered out, practically represent the sum of the two first terms of Eq. 1. We subtracted the annual values of the past long-term variations ($S_L$; see Sect. 3.1) from the observed time series, to obtain the short-term variability $S_S$.

We then calculated the ccdf for the short-term sea level variations. We extrapolated the ccdf to frequencies of exceedance

10 smaller than 1/46 years with an exponential function fitted to the tail of the ccdf (Fig. 4). The exponential function was fitted to sea levels with a frequency of exceedance of 5 events/year or less. Särkkä et al. (2017) examined different functions and methods for extrapolating sea level ccdfs at Helsinki. They found that both a Weibull and an exponential extrapolation of simulated daily sea level maxima produced results well in line with a Generalized Extreme Value (GEV) fit to annual simulated sea level maxima.





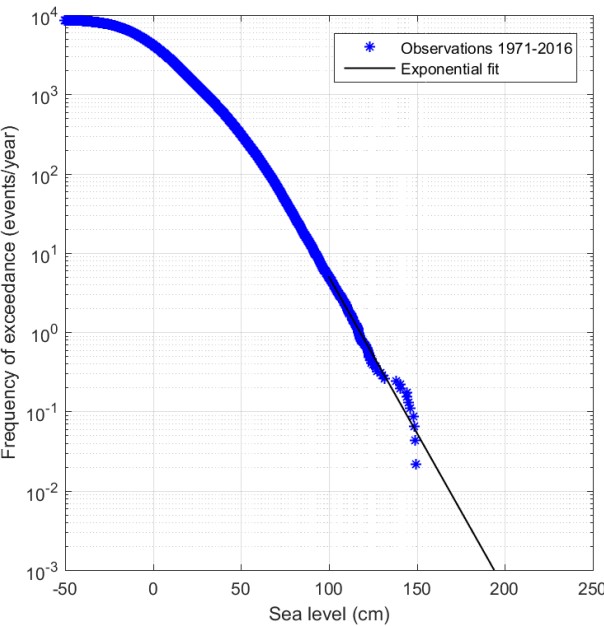

**Figure 4.** Ccdf of the short-term sea level variations at the Helsinki tide gauge: observed hourly values in 1971–2016, from which a time-dependent estimate for the long-term mean sea level has been subtracted.

### 4.3 Distributions of the wind wave run-up

The short time series measured at Jätkäsaari and Länsikari (Sect. 3.3) are not long enough for constructing the local wave height probability distributions. We therefore compared these measurements to the simultaneous open sea data from the Gulf of Finland to determine an attenuation factor for each wave direction and wave period. The attenuation factors were then applied to the 15-year open sea measurement record to produce estimates of the wave conditions at the study locations. We calculated hourly significant wave heights from two consecutive measured 30-minute values, to be able to combine these with the hourly sea level data.

The wave height values obtained by attenuating the open sea data were combined with the local measurements, and ccdfs were estimated by fitting piecewise exponential functions to the data. For the large values of the ccdf the exponential function was fitted to the observational data, while for the smaller values (rarer events) a fit was made to the modelled values. These two pieces were connected to form one continuous distribution. The distribution was then extrapolated using an exponential fit. Since neither the observations nor the modelled values are by themselves sufficient to form a probability distribution, the above method was chosen to make the most efficient use of both data sets.

The final step was to estimate the wave run-up, i.e. the maximum vertical elevation of the water in relation to the still water level during a certain period, which in our case was an hour. The wave run-up can be calculated for different percentages, e.g.



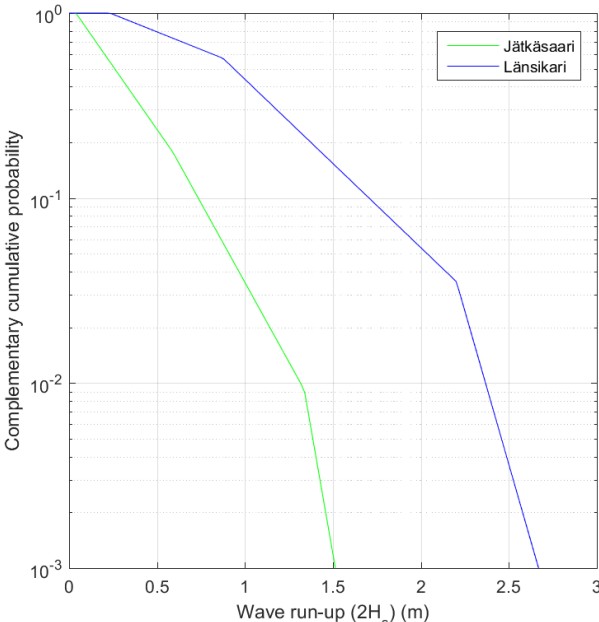

**Figure 5.** Wave run-up distributions for the two locations in the Helsinki archipelago: Länsikari and Jätkäsaari.

as the water level exceeded 2% of the time. We set out to seek a conservative estimate for the level exceeded once during the one hour time period.

The run-up height depends on a number of parameters, but on a steep, sufficiently deep shoreline, the maximum vertical elevation is determined by the highest single individual wave, which is further magnified by reflection. As a simplification, we transformed the significant wave height, $H_s$, to the maximum single wave, $H_{max}$, by a multiplication by two. This is an upper limit compared to the widely used Rayleigh distributions and empirical distributions (Forristal, 1978). We also assumed that the highest single wave is completely reflected, which is a conservative estimate based on direct measurements of reflected waves at a steep coastal construction (Björkqvist et al., 2017c); this resulted in another multiplication of $H_{max}$ by two to get the maximum reflected wave height. Finally, the height of the crest of such wave above the still water level is half of the wave height, which is defined from trough to crest. Thus: $H_{runup} = 2H_s$. The resulting cumulative wave run-up distributions are illustrated in Fig. 5.

## 4.4 Probability of the sum of two independent random variables

The theory for determining the sum of two random variables can be found in text books (e.g. Schay, 2016), but it will nonetheless be outlined below for completeness and to introduce notation.





Let $\underline{x}, \underline{y}, \underline{z} \in \mathbb{R}$ be continuous random variables, which can take values denoted by $x$, $y$ and $z$ respectively. We use the established notation of $f_x, f_y, f_z$ and $F_x, F_y, F_z$ for the associated probability density functions and cumulative distribution functions (Eq. 2). We now define $\underline{z} := \underline{x} + \underline{y}$ to be the sum of the independent random variables $\underline{x}$ and $\underline{y}$, while imposing no further constraints on $\underline{x}$ or $\underline{y}$.

The goal is to define the cumulative distribution function $F_z$, namely expressing the probability $P\{\underline{z} \leq z\}$ for an arbitrary $z \in \mathbb{R}$. As $\underline{z}$ is given as the sum $\underline{x} + \underline{y}$, it's easy to realise that $\underline{z} = z$ when $\underline{x} = \xi$ and $\underline{y} = z - \xi$ for any $\xi \in \mathbb{R}$. Consequently, $\underline{z} \leq z$ when $\underline{x} = \xi$ and $\underline{y} \leq z - \xi$, since $\underline{z} := \underline{x} + \underline{y} \leq \xi + (z - \xi) = z$. By using the assumption of independence the probability of the occurrence can be expressed as a product, thus yielding

$$
\begin{aligned}
P\{(\underline{x} = \xi) \wedge (\underline{y} \leq z - \xi)\} &= P\{\underline{x} = \xi\} \cdot P\{\underline{y} \leq z - \xi\} \\
&= f_x(\xi) \cdot F_y(z - \xi).
\end{aligned}
$$

Since this holds for any $\xi \in \mathbb{R}$ and the probability $P\{\underline{z} \leq z\}$ is a sum of all these occurrences, we can express $F_z(z)$ as the convolutions integral

$$
F_z(z) = P\{\underline{z} \leq z\} = \int_{\mathbb{R}} f_x(\xi) F_y(z - \xi) \, \mathrm{d}\xi = f_x * F_y. \tag{3}
$$

For practical purposes $f_x$ and $F_y$ are usually given as discrete functions. By defining the discrete functions as

$f_x, F_y, F_z : \{i = n \cdot \Delta \xi \mid n \in \mathbb{Z}\} \rightarrow [0, 1]$

for some $\Delta \xi \in \mathbb{R}$ and redefining $f_x$ as the probability mass function fulfilling $\sum_i f_x(i) = 1$, we end up with the discrete version of Eq. 3:

$$
F_z(z) = \sum_{i=-\infty}^{\infty} f_x(i) F_y(z - i). \tag{4}
$$

### 4.5   Distributions of the sum of sea level variations and wind waves

We applied the method for calculating the probability of the sum of two random variables (Sect. 4.4) to get the probability distribution of the sum of the three factors (Eq. 1) from the probability distributions of each of those (Sect. 4.1, 4.2, and 4.3). As the first step, we calculated the cdf of the still water level ($S_L + S_S$). This distribution accounts for the sea level variations only and will be referred to as the **SL-distribution**.

For the present conditions (year 2017), we calculated the SL-distribution simply by adding the long-term mean sea level
estimate of 18.7 cm (in the N2000 height system) to the values for the short-term sea level variability. For the future (years 2050 and 2100), we calculated the SL-distribution as the convolution $F_{SL} = f_{S_L} * F_{S_S}$ of the pdf of the long-term mean sea level scenarios ($S_L$) and the cdf of the short-term sea level variability ($S_S$). Still water levels corresponding to certain frequencies of exceedance are shown in Table 1.





**Table 1.** Still water levels (in m relative to N2000) corresponding to certain frequencies of exceedance for three years (2017, 2050 and 2100) based on the observed sea level variability and mean sea level scenarios for the Helsinki tide gauge.

| | Helsinki tide gauge | | |
| --- | --- | --- | --- |
| | Prediction year | | |
| | **2017** | **2050** | **2100** |
| Frequency of exceedance (events/year) | Still water level (m) | | |
| 1/1 | 1.36 | 1.49 | 2.33 |
| 1/50 | 1.80 | 1.92 | 2.87 |
| 1/100 | 1.87 | 2.00 | 2.95 |
| 1/250 | 1.97 | 2.10 | 3.06 |

As a second step, we calculated the cdf of the full three-component sum (Eq. 1). By using the notations from Sect. 4.4, $\underline{x}$ is the still water level $S_L + S_S$, $\underline{y}$ is the run-up $H_{runup}$, and $\underline{z}$ is the elevation to which the continuous water mass reaches, $H$. Since the method is symmetric, the choice of $\underline{x}$ and $\underline{y}$ is in theory arbitrary. In practice, more data are required to get a good estimate of the pdf $f_x$, which guides the proper choice of variables. We had significantly more sea level data available and will

for the remainder of this paper adopt the notation $f_{SL}$ ("sea level") and $F_W$ ("wave") for $f_x$ and $F_y$ in Eq. 4. The combined cumulative function obtained using convolution and corresponding to $F_z$ in Eq. 4 will be denoted $F_{SL,W} = f_{SL} * F_W$. The resulting distribution will be referred to as the **SL,W-distribution**.

## 5 Sensitivity of the method on the shape of the wave distribution

The wave run-up distributions used in this study are experimental fits to the available wave data at different locations. For

this, we wanted to test more generally how different wave height conditions influence the resulting joint effect of sea level and waves when the sea level distribution is kept unchanged. We therefore constructed six different theoretical wave run-up distributions with known parameters to quantify the effects of different properties of the wave distributions (shape, expected value and typical magnitude relative to the sea level variations) on the distribution of the sum of waves and sea level.

### 5.1 Setup of the sensitivity analysis

The theoretical wave run-up distributions were constructed as two-parameter Weibull distributions, which have probability functions (pdfs and cdfs)

$$f(x, k, \lambda) = \frac{k}{\lambda} \left( \frac{x}{\lambda} \right)^{k-1} \exp \left( -\frac{x}{\lambda} \right)^k \tag{5}$$





**Table 2.** The different theoretical wave run-up distributions and observation based still water level distribution used for the sensitivity test. The Weibull scale parameter ($\lambda$), shape parameter ($k$), expected value $\mathbb{E}$, $90^{th}$, $95^{th}$ and $99^{th}$ percentiles are given for the wave run-up distributions, and the same percentile values for the still water level distribution.

| Distribution | $\lambda$ | $k$ | $\mathbb{E}$ | $90^{th}$ perc. | $95^{th}$ perc. | $99^{th}$ perc. |
|---|---|---|---|---|---|---|
| Wave1a | 0.2 | 2.0 | 0.18 m | 0.30 m | 0.35 m | 0.43 m |
| Wave1b | 0.2 | 2.5 | 0.18 m | 0.28 m | 0.31 m | 0.37 m |
| Wave2a | 0.5 | 2.0 | 0.44 m | 0.76 m | 0.87 m | 1.07 m |
| Wave2b | 0.5 | 2.5 | 0.44 m | 0.70 m | 0.78 m | 0.92 m |
| Wave3a | 1.5 | 2.0 | 1.33 m | 2.28 m | 2.60 m | 3.22 m |
| Wave3b | 1.5 | 2.5 | 1.33 m | 2.09 m | 2.33 m | 2.76 m |
| Still water level | - | - | 0.00 m | 0.33 m | 0.45 m | 0.68 m |

$$F(x, k, \lambda) = 1 - \exp\left(-\frac{x}{\lambda}\right)^k, \tag{6}$$

where $k$ is the shape parameter and $\lambda$ is the scale parameter. Our six wave height distributions form three pairs, where each pair has an equal expected value, but slightly different shape parameters $k$ (Table 2).

The three wave run-up pairs represent three different wave conditions. The first pair (W1a and W1b) represents a typical
sheltered situation where the wave height is small in comparison to the more dominant sea level variations. For the second pair (W2a and W2b) the waves and the sea level variations are of similar magnitude, while the third pair (W3a and W3b) represents waves that are clearly dominant compared to the sea level variations. The sea level distribution used in the sensitivity analysis was based on 46 years of observations (see Sect. 4.2). We call the sea level distribution the still water level distribution SL and denote the wave height distributions by wave run-up distributions W1a, W1b etc.

An overview of the different distributions, their parameters and properties is found in Table 2. The probability density functions are plotted in Fig. 6, which gives a visual comparison of the different wave height situations in relation to the sea level variations. The effect of the slightly larger shape parameter of distributions W1b, W2b and W3b compared to W1a, W2a and W3a can be seen as a slightly narrower and sharper form of the wave height distributions.

We performed all the calculations using the wave height cdfs and the sea level pdf, as reasoned above in Sect. 4.5. Wave
run-up heights were summed with the still water levels. The distribution of the sum of these two variables i.e. the run-up level distribution is labeled SL,W1a, where $F_{SL,W1a} = f_{SL} * F_{W1a}$ (see Eq. 3).

## 5.2  Comparison of the test outcomes and their relation to a theoretical framework

We chose four different frequencies of exceedance (1/1, 1/50, 1/100 and 1/250 events/year) for a closer examination. Table 3 summarizes these for the still water level distribution and the six wave run-up distributions, as well as for the sum of these i.e.



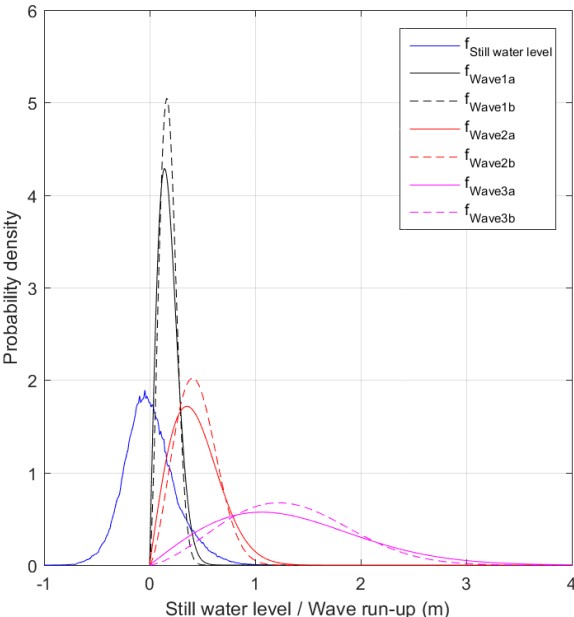

**Figure 6.** The probability density functions of the still water level distribution and the six different wave run-up distributions.

the run-up level distributions. As a comparison, also the corresponding still water levels added to the expected values of the wave run-up distributions are shown.

For the first pair, the sea level variations are clearly dominating the wave variations (W1a and W1b, see Table 2). The run-up levels are mostly set by the still water levels; the distribution of the sum of the sea level and the wave run-up produces about 0.2

5  m higher values in comparison to the still water levels alone. In this setting, the effect of different shapes of the wave run-up distributions W1a and W1b on the results is negligible.

In the second pair, neither the wind waves nor the sea level variations are clearly dominant (W2a and W2b, see Table 2). The contribution of the waves is now significant. Even the run-up level with a frequency of 1/1 events/year for the SL,W2a (1.95 m) is larger than the still water level with a frequency of 1/250 events/year (1.79 m). The effect of the shapes of the wave

10  distributions is no longer negligible for the second pair. The distribution W2a has a thicker tail compared to distribution W2b, meaning that the higher values are more probable. The difference in the run-up level with a frequency of exceedance of 1/250 events/year between SL,W2a and SL,W2b is already 0.2 m (Table 3). This difference is caused solely by the different shapes of the wave distributions W2a and W2b, since they have the same expected value and the sea level distribution was identical in both cases.

15  In the case of the third pair, the contribution of the larger waves becomes evident. The run-up levels of the sum of sea level and waves are up to 3.5 m (1/1 events/year) and 4.1 m (1/250 events/year) higher compared to the still water levels alone. There is a 0.6 m difference in still water level when comparing the frequencies 1/1 events/year and 1/250 events/year, but the increase





in the corresponding run-up levels is up to 1.3 m (Table 3). Unlike for the previous cases, the effect of the wave distribution's shape factor on the run-up level increases with smaller frequencies of exceedance, and is up to 1.2 m for frequency of 1/250 events/year. This shape related behavior can be seen explicitly in Fig. 7.

The sensitivity test reveals that the dominant component determines strongly the run-up level of the distribution of the sum. In addition to the remarks above, this effect is evident as the still water levels are rather close to the run-up levels SL,W1a and SL,W1b (the case where sea level variations are dominant), and on the other hand as the third wave run-up pair W3a and W3b constitutes mostly the run-up levels SL,W3a and SL,W3b (the case where waves dominate over sea level variations) (Table 3).

The distribution of the run-up levels is determined by the relative magnitude of the components, which is in accordance with the properties of the probability of the sum of two independent random variables. The expected value of the sum of two such

variables is the sum of the expected values of the components, and the variance is the sum of the variances of the individual components. For this reason, the larger component dominates strongly in the standard deviation of the sum:

$$\sigma = \sqrt{\sigma_{SL}{}^2 + \sigma_W{}^2}, \tag{7}$$

where $\sigma_{SL}$ is the standard deviation of the still water level distribution and $\sigma_W$ is the standard deviation of the wave run-up distribution. Since the mean of the still water level distribution is zero (see Table 2) the expected value of the distribution of

the sum will be determined by the waves. The run-up level $z$ corresponding to the cumulative probability $P$ can therefore be written as

$$z(P) = \mathbb{E}(W) + \xi_{SL,W}(P) \cdot \sqrt{\sigma_{SL}{}^2 + \sigma_W{}^2}, \tag{8}$$

where $\xi(P)$ is a coefficient depending on the shape of the distribution. When the sea level variations dominate, i.e. $\sigma_{SL} \gg \sigma_W$, the shape of the distribution is also mainly determined by the still water level distribution, and we can approximate Eq. 8

by

$$z(P) \approx \mathbb{E}(W) + \xi_{SL}(P) \cdot \sigma_{SL}, \tag{9}$$

where $\mathbb{E}(W)$ takes the role of a "fixed wave action height". Equation 9 shows that $\mathbb{E}(W)$ is only "fixed" at locations with similar wave conditions. As long as the sea level variations dominate ($\sigma_{SL} \gg \sigma_W$), and the coefficient $\xi$ depends on the still water level distribution, there is no need to calculate the distribution of the sum to define the run-up level. However, if $\sigma_W$ is

not negligible compared to $\sigma_{SL}$, the coefficient $\xi_{SL,W}$ will differ from $\xi_{SL}$. The approximation in Eq. 9 will no longer be valid and the full distribution of the sum needs to be identified in order to account for the waves.

If both probability distributions under investigation are normal (Gaussian) distributions, the distribution of the sum is also Gaussian. In this case, $\xi(P)$ is similar for each of them, and we get the simple equation (9) for the run-up level. Weibull and exponential distributions are sufficiently close to the normal distribution that we can expect that also in their case the run-up

level can in practice be defined by using this simple form, provided that still water level variations dominate sufficiently.



In our sensitivity test (Table 3) the run-up levels of the sum of sea level and waves (SL,W1a and SL,W1b) differed at most 0.1 m from the run-up levels based on the sum of still water levels and expected values of the wave run-up distributions, namely SL+𝔼(W1a) and SL+𝔼(W1b). Thus, in a situation where the sea level variations dominate, simply adding the expected value of the wave run-up distribution on top of the still water levels produces results quite similar to those based on the distribution

of the sum. In this situation, the simplified equation 9 works pretty well for the Weibull distributed random variables used in the sensitivity test.

However, as soon as the contribution of the waves increases, the situation changes. In the equal wave-sea level situation (SL,W2a and SL,W2b vs. SL+𝔼(W2a) and SL+𝔼(W2b)), simply adding the expected value of the wave run-up distribution on top of the still water levels would underestimate the run-up levels by up to 0.4 m compared to the distribution of the sum.

However, when looking at the difference between the still water levels and the run-up levels of the sum of sea level and waves, we notice that the effect of the waves can be quantified almost as a constant value to be added on top of the still water levels for all the four frequencies of exceedance under inspection.

Finally, a similar comparison for the case where the waves dominate (SL,W3a and SL,W3b vs. SL+𝔼(W3a) and SL+𝔼(W3b)) results in significant differences (up to 2.8 m), showing that the simplified solution of adding the expected value of the wave

run-up distribution on top of the still water levels would lead to a significant underestimation of the run-up level. Moreover, it is clear that for this situation the effect of the waves cannot be quantified as a constant value to be added on top of the still water levels for the frequencies of exceedance ranging from 1/1 to 1/250 events/year.

## 6 Results of the case study in Helsinki Archipelago

We applied the presented method in the Helsinki Archipelago, located at the northern coast of the Gulf of Finland, Baltic Sea.

The calculations were done for two locations, where Jätkäsaari is situated deep inside the archipelago near the shoreline, while Länsikari is more exposed to the open sea wave conditions (Fig. 2).

We calculated the SL-distributions for the still water level as a sum of two components: the short- and long-term sea level variations. In the SL,W-distributions, the wave run-up was additionally accounted for, as they were calculated as a sum of three components as outlined in Sect. 4. These distributions are illustrated in Fig. 8. We calculated the distributions both for

the present conditions (2017), and for the future scenarios in 2050 and 2100. The run-up levels representing the maximum elevation of the continuous water mass on a steep shore with selected frequencies of exceedance are given in Table 4.

The run-up levels for a location closer to the open sea (Länsikari) are up to 1.2 m higher compared to the values for the sheltered shore location (Jätkäsaari). This clear difference follows from the difference in the wave run-up distributions (see Fig. 5), and highlights the variability of the waves due to locational differences, even in a rather small coastal area under

investigation.

The sensitivity test (Sect. 5) showed that the dominant component determines strongly the run-up level of the distribution of the sum, which results from the properties of the probability of the sum of two independent random variables. For the sheltered





**Table 3.** Sensitivity test results for different frequencies of exceedance for the still water level distribution SL, the six theoretical wave run-up distributions W1a-W3b, and the run-up level distributions SL,W1a, i.e. the convolution $f_{SL} * F_{W1a}$. The run-up levels resulting from the sum of still water level and expected value of the wave run-up distributions are marked by SL+$\mathbb{E}$(W1a).

| | Frequency of exceedance (events/year) | | | |
|---|---|---|---|---|
| Distributions | 1/1 | 1/50 | 1/100 | 1/250 |
| | Still water level (m) | | | |
| SL | 1.18 | 1.61 | 1.69 | 1.79 |
| | Wave run-up (m) | | | |
| W1a | 0.60 | 0.72 | 0.74 | 0.76 |
| W1b | 0.48 | 0.56 | 0.57 | 0.59 |
| W2a | 1.51 | 1.80 | 1.85 | 1.91 |
| W2b | 1.21 | 1.39 | 1.42 | 1.46 |
| W3a | 4.52 | 5.41 | 5.55 | 5.73 |
| W3b | 3.63 | 4.18 | 4.27 | 4.38 |
| | Run-up level (m) | | | |
| SL,W1a | 1.40 | 1.83 | 1.91 | 2.01 |
| SL,W1b | 1.38 | 1.81 | 1.89 | 1.99 |
| SL,W2a | 1.95 | 2.40 | 2.48 | 2.58 |
| SL,W2b | 1.81 | 2.24 | 2.32 | 2.42 |
| SL,W3a | 4.66 | 5.58 | 5.73 | 5.92 |
| SL,W3b | 3.84 | 4.48 | 4.58 | 4.71 |
| SL+$\mathbb{E}$(W1a) | 1.35 | 1.79 | 1.86 | 1.96 |
| SL+$\mathbb{E}$(W1b) | 1.35 | 1.79 | 1.86 | 1.96 |
| SL+$\mathbb{E}$(W2a) | 1.62 | 2.05 | 2.13 | 2.23 |
| SL+$\mathbb{E}$(W2b) | 1.62 | 2.05 | 2.13 | 2.23 |
| SL+$\mathbb{E}$(W3a) | 2.51 | 2.94 | 3.01 | 3.12 |
| SL+$\mathbb{E}$(W3b) | 2.51 | 2.94 | 3.02 | 3.12 |

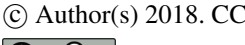



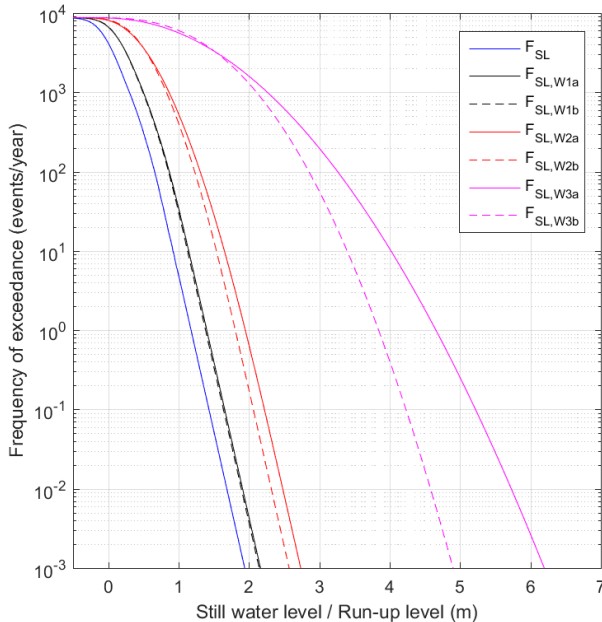

**Figure 7.** The cumulative probability distribution functions for the observation-based sea level variability and the sum of sea level and waves, obtained by applying six different theoretical significant wave height distributions.

shore location (Jätkäsaari) the run-up is mostly set by the still water level (Table 1), whereas for the study location exposed to open sea (Länsikari) the contribution of waves becomes more significant.

The sensitivity test also revealed that in a situation where the sea level variability dominates, the effect of the waves could be quantified as a constant addition on top of the still water levels exceeded with different frequencies, but not for the case

where the wave run-up was clearly dominant in comparison to the sea level variations. The case study results for 2017 and 2050 show the same effect, as the difference between the still water levels in the SL-distributions and the run-up levels in the SL,W-distributions is almost independent of the frequency of exceedance (Tables 1 & 4). This difference – representing the effect of the wave run-up – varies only up to 6 cm between the frequencies of exceedance from 1/1 to 1/250 events/year. Even in the 2100 scenario, this difference only varies up to 17 cm, a small value compared to the total effect of the waves which is

of the order of 1–2 m.

The impact of the future mean sea level change is evident in the SL-distributions for the three different years (Fig. 8). The still water levels corresponding to certain frequencies of exceedance change only slightly from 2017 to 2050, but increase significantly more from 2050 to 2100. From 2050 to 2100, the 1/1 events/year still water level increases by 84 cm, and the 1/250 events/year still water level by 96 cm (Table 1). This change results from the predicted accelerating mean sea level rise

in the Gulf of Finland, as well as from the wider uncertainty range in the mean sea level projections for 2100, which is reflected in the mean sea level probability distribution (for details, see Pellikka et al., 2017).



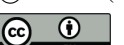

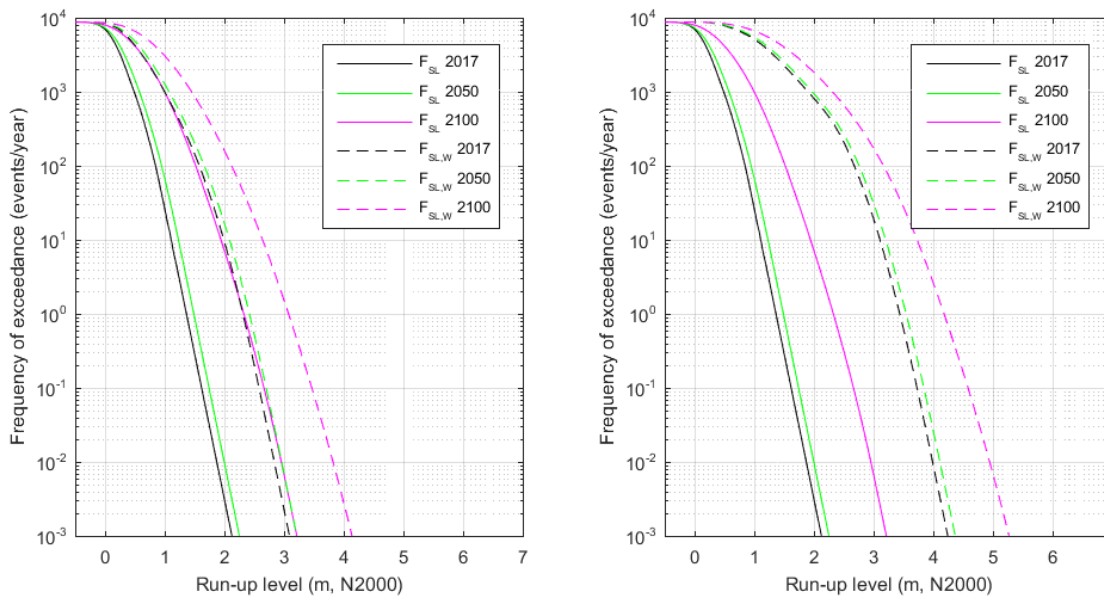

**Figure 8.** Ccdfs for the still water level alone ($F_{SL}$), and the sum of the still water level and wave run-up ($F_{SL,W}$) for three different years (2017, 2050 and 2100) at the two case study locations: Jätkäsaari (on the left) and Länsikari (on the right).

**Table 4.** Run-up levels (m relative to N2000), as the sum of still water level and wave run-up, for three different years (2017, 2050 and 2100) for Jätkäsaari and Länsikari.

| | Jätkäsaari | | | Länsikari | | |
|---|---|---|---|---|---|---|
| | Prediction year | | | Prediction year | | |
| | **2017** | **2050** | **2100** | **2017** | **2050** | **2100** |
| Frequency of exceedance (events/year) | Run-up level (m) | | | Run-up level (m) | | |
| 1/1 | 2.31 | 2.42 | 3.07 | 3.44 | 3.54 | 4.18 |
| 1/50 | 2.76 | 2.88 | 3.72 | 3.91 | 4.03 | 4.84 |
| 1/100 | 2.84 | 2.96 | 3.82 | 3.99 | 4.11 | 4.94 |
| 1/250 | 2.94 | 3.06 | 3.95 | 4.09 | 4.21 | 5.08 |

As we used the same mean sea level scenario for both Jätkäsaari and Länsikari, the effect of the mean sea level change is similar for them even in the SL,W-distributions. For example, the run-up levels exceeded by 1/100 events/year increase by 86 cm in Jätkäsaari and 83 cm in Länsikari from 2050 to 2100. The small difference between the two study locations results from the slightly different shape of the wave run-up distributions.



## 7    Discussion

### 7.1    Conditions and applicability of the method

In general case, the relationships between the wave height, wave run-up, and sea level variations are complex. In this study, we made several assumptions and simplifications. The aim of this section is to discuss the validity of our results, and also help the reader to estimate whether this method could be used in a certain location or with a specific data available.

The essential prerequisites for applying the method presented above are:

1. An estimate for the long-term mean sea level is needed. In its simplest form, this can be a single mean sea level height value. If the mean sea level is changing, however, an estimate for this change is needed. Again, a simple estimate could be a time-dependent mean sea level value; a linear trend, for instance. Using an ensemble of estimates, the way we did with the future scenarios, however, leads to a time-dependent probability distribution for the mean sea level, which contains more information on the different possible future pathways.

2. An estimate for the range of the short-term sea level variability is needed; technically, in the form of a good-quality probability density function. In the case of the Finnish coast, we have found that several decades of observations with hourly time resolution are needed to get a reliable estimate for the extent of the local sea level variations. Additionally, to estimate run-up levels with low frequencies of exceedance, such as 1/250 events/year used in this study, the observation-based probability distribution – rarely extending down to frequencies below 1/100 events/year – needs to be extrapolated using suitable extreme value analysis methods.

3. An estimate for the wave run-up distribution is needed to account for the effect of waves on the coast. In this paper we have used the simplest formula for a steep shore using the highest single wave, which was estimated from the significant wave height $H_s$. The method can be generalised by using wave run-up formulations that also account for e.g. the slope of the beach.

4. We based our analysis on a simplifying assumption that the sea level variations and wave run-up are independent. This makes it possible to calculate the distribution of the sum from the marginal distributions without additional assumptions. In practice, the independence of the variables can be, at least partly, achieved for locations with a constant beach profile, such as deep and steep shores. Strong wind-independent components in the sea level also decrease the dependence of the sea level and the wave run-up. In the Baltic Sea, such component is the total Baltic Sea water volume which, although expressing a strong correlation with the wind conditions (Johansson et al., 2014), does so in a time scale much longer than that of the wind waves. In addition, the mutual dependence of the sea level and waves is weakened in the Gulf of Finland, since strong easterly winds lower the sea level by emptying the gulf. Tidal variations are also a sea level component which is independent of waves; such variations are small on the Finnish coast, however.

As long as the above conditions are met, we consider the method presented here applicable also for other places than the Finnish coast. Naturally, as the most important factors causing sea level variations are different in different places, this needs




to be taken into account. For instance, in places where the tidal variations dominate over storm surges, a different analysis of the short-term sea level variability might be appropriate.

## 7.2 Limitations and potential improvements

In our approach, we treated the still water level variations and the wave run-up as independent variables as a first approximation. The limited amount of wave data available for this study imposed challenges in the construction of the full joint distribution, which would have taken into account the possible dependencies between these variables. The dependency might be affected by the location specific circumstances, and further studies are needed to determine the conditions under which the use of the full two dimensional distributions is preferable to assuming independence.

Block maxima – such as monthly maxima – have often been used for the extreme value analysis of sea level variations. However, this implicitly restricts the study of the joint effect to cases where the still water level is high, thus excluding combinations of a moderate still water level and high waves. The impact on the end result is therefore expected to be influenced by the relative importance of the two phenomena.

Pellikka et al. (2017) used the observed monthly maxima of sea levels on the Finnish coast to calculate the location-specific short-term sea level variability distributions. They calculated the probability distribution of the sum of long- and short-term sea level variations with a method similar to the one we used to calculate the SL-distribution. By this method they analyzed the present and future flooding risks on the Finnish coast. Our results for still water levels with frequencies of exceedance of 1/1, 1/50 and 1/100 events/year (Table 1) are higher than those of Pellikka et al. (2017). This is mostly explained by the differences in statistics. Several high hourly sea level values can occur during the same month, or even the same storm surge event, and still result into only one monthly maximum in the statistics. Thus, the hourly values have a higher frequency of exceedance than the monthly maxima, reflecting the difference in the definition of "an event" in each case.

We calculated the future scenarios for the flooding risks by simply combining the mean sea level scenarios with the present-day short-term sea level variability and wave conditions. Thus, we implicitly assumed that those will not change in the future. A potential improvement, to get deeper insight into the changes of flooding risks in the future, would be to include scenarios of short-term sea level variability or wave conditions. As these both mainly depend on short-term weather (wind and air pressure) conditions, this would require scenarios for the short-term weather variability.

## 8 Conclusions

In this study, a location-specific statistical method was used for the first time on the Finnish coast for evaluating flooding risks based on the joint effect of three different components: 1) long-term mean sea level change, 2) short-term sea level variability, and 3) wind-generated waves. We conducted an observation-based case study for two locations with steep shorelines, and performed a sensitivity test with theoretical wave run-up distributions. The probability distributions of the sum of the three aforementioned components were calculated, giving the elevations to which the continuous water mass can rise as a result of




still water level and the wind wave run-up. Such probability distribution provides direct run-up level estimates for different frequencies of exceedance, which enables an easy evaluation of different risk levels for coastal building.

The case study at the Helsinki Archipelago (Sect. 6) showed that the joint run-up levels for a location exposed to the open sea (Länsikari) are clearly higher compared to the values for the sheltered location near the shoreline (Jätkäsaari). This finding

supports the need for a location-specific evaluation of the wave height to prevent over- and underestimation of the joint effect, especially in places with an irregular coastline.

The effect of the mean sea level scenario on the run-up levels at different frequencies of exceedance at the case study sites is moderate in 2050, but more prominent in 2100. This is due to the predicted acceleration of the mean sea level rise as well as the increasing uncertainties in the future, resulting in a wider probability distribution. Pellikka et al. (2017) have discussed these

scenarios in more detail. According to the case study presented here, the coastal flooding risks including the simultaneous effect of sea level and waves are significantly higher in the end of the century compared to the current state. However, the run-up levels estimated for 2100 do not form the expected run-up level distribution for that year, but include the uncertainty of the future mean sea level rise. Thus, the run-up levels with certain frequency of exceedance in 2100 are just statistical estimates accounting for all the possible mean sea level scenarios, only one of which will eventually be realized in 2100.

Safe coastal building elevations are usually estimated for structures with a designed lifetime of at least several decades, but the relevant safety margins differ between commercial buildings, residential buildings and e.g. nuclear power plant sites. We therefore need to consider scenarios up to 2100 and frequencies of exceedance as rare as 1/250 events/year or even less. The approach presented in this paper allows the evaluation of separate risk levels for different coastal infrastructures, and thereby assists in a cost-effective coastal planning to meet the requirements of changing climate of the future.

*Competing interests.*   No competing interests are present.

*Acknowledgements.*  This research was partly funded by: the Finnish State Nuclear Waste Management Fund (VYR) through SAFIR2018 (the Finnish Research Programme on Nuclear Power Plant Safety 2015-2108); The City of Helsinki, and Arvid och Greta Olins fond (Svenska kulturfonden, 15/0334-1505).



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
