# Peer review of "Combining probability distributions of sea level variations and wave run-up to evaluate coastal flooding risks"

_Natural Hazards and Earth System Sciences, 2017_

## Referee Comment (RC1) · K.-D. Suh (Referee) · 1 Feb 2018

Review report on "Combining probability distributions of sea level variations and wave run-up to evaluate coastal flooding risks" by Ulpa Leijala et al.

Major points:

1.  In page 9, it is written: "The exponential function was fitted to sea levels with a frequency of exceedance of 5 events/year or less." Why? The frequency of exceedance of the observed data in Figure 4 is from 1/46 to about 8000 events/year. It is thought that only the data of low frequency of exceedance are used in the curve fitting because we are interested in the events of high sea level. The reason why the data of low frequency of exceedance are used should be explained.

2.  In Figure 4, the maximum frequency of exceedance occurs at the sea level of -50 cm, indicating that negative storm surges frequently occur in the study area. The reason for this should be explained in the paper.

3.  In page 10-11, it is written: "The wave run-up can be calculated for different percentages, e.g. as the water level exceeded 2% of the time. We set out to seek a conservative estimate for the level exceeded once during the one hour time period." In the design of coastal defense structures, it is common to use the 2% run-up height to determine the crest freeboard. If the mean wave period is 8 s, the wave run-up exceeding 2% run-up height occurs 9 times during one hour, whereas the run-up height exceeded only once during one hour is exceeded 0.22% of the time. Therefore, taking the run-up height exceeded once during the one hour time period is too conservative from the engineering point of view.

4.  In page 11, the relationship $H_{max} = 2H_s$ is used. Longuet-Higgins (1952, J. Marine Res. 11, 246-266) presented the relationship $H_{max} = 0.707\sqrt{\ln N}H_s$ for a storm with a relatively large number of waves $N$. Again, if the mean wave period is 8 s, the number of waves during one hour is 450, which gives $H_{max} = 1.75H_s$. Therefore, the relationship $H_{max} = 2H_s$ may be too conservative.

5.  The assumption of complete wave reflection from a coastal structure (i.e. $H_{runup} = H_{max}$) may also be a too conservative assumption. This assumption, however, could be justified if we take into account the effect of wave nonlinearity in shallow water (i.e. peaked crest and flat trough), which was not considered in this study.

6.  Sorensen (2006, Basic Coastal Engineering, 3rd ed., Springer, p. 237) presented the relationship $R_p = R_s\sqrt{\ln(1/p)/2}$ where $R_p$ is the wave run-up height of the exceedance probability $p$ and $R_s$ is the run-up height of the incident significant wave height as if it were a monochromatic wave. If we use $p = 0.02$ and $R_s = H_s$ (i.e. complete wave reflection), $H_{runup} = R_{2\%} = 1.4H_s$ which is 70% of the value used in this study. On the other hand, if we use $p = 0.0022$, which is the exceedance probability of the wave height exceeded only once during one hour (when the mean wave period is 8 s), $H_{runup} = R_{0.22\%} = 1.75H_s$. This changes to $H_{runup} = H_{max}$ (using the relationship $H_{max} = 0.707\sqrt{\ln N}H_s$), which is the same as the run-up height used in this study except that $H_{max}$ is not calculated as $2H_s$ but as $1.75H_s$. In conclusion, to avoid too conservative estimate for wave run-up height, either $H_{runup} = 1.4H_s$ (general design standard) or $H_{runup} = 1.75H_s$ (run-up height exceeded once during one hour as taken in this study) should be used.

7. In addition to Table 1, it may be worthwhile to show the curves of $F_{SL}$ for 2017, 2050, and 2100.
8. Two-parameter Weibull distributions are used for the sensitivity analysis. It may be better to add the fitted Weibull distributions (along with the shape and scale parameters) in Figure 5 to show that the Weibull distribution fits well the observation.

Minor points:
1. 1$^{st}$ line below Eq. (1): wave height >> wave run-up height

---

## Referee Comment (RC2) · J. A. Jiménez (Referee) · 14 Feb 2018

Review of ***Combining probability distributions of sea level variations and wave run-up to evaluate coastal flooding risks***

Manuscript **# nhess-2017-438**

The manuscript presents a joint probability analysis of sea level – wave induced runup in the Finish coast to be used in coastal flooding assessment. In this sense, the topic is on the scope of NHESS and can be of interest for NHESS readers. In what follows, some comments are given.

**[1]** The first comment is purely formal. Authors state in lines 22-23 (pag 3) that they are going to call "run-up level" to the combined water elevation (mean water level and wave run-up contributions. This is misleading since it is not the unusual approach in the literature. It should be better to use something like "total water level" to avoid confusion with the standard wave-induced run-up.

**[2]** *Lines 13-14 (pag 3).* Coastal floods are also a consequence of storm-surges. Please rephrase the sentence.

**[3]** *Lines 6 (pag 4).* In general terms, the wave-induced component of the water level at the shoreline is the run-up and not the wave height (a different thing is that you approach the run-up with the wave height but this depends on how you calculate it).

**[4]** *Line 1 (pag 5).* Long-term mean sea level does not change from decade to decade. Mean sea level is continuously varying and "long-term" refers to the low-frequency component which, apparently, you consider to be associated to periods in the scale of decades (or longer).

**[5]** *Section 3.1. Long-term sea level.* You are using long-term estimations of sea level at selected horizons based on a paper that is under review. If this component is important for your calculations, it can be difficult for some readers to trust on it without having access to the scientific work supporting used values.

**[6]** *Section 3.2.* Please change the heading. Here you are not describing variability but just the existing data. They are simply water level measurements acquired by using tidal gauges. Use something similar to heading of section 3.3 (e.g. tidal data, water level data).

**[7]** *Lines 6-7 (Page 8).* Please remove the last sentence "The significant wave height is …". If you want to use a definition of Hs use a formal one (e.g. based on spectral moments).

**[8]** *Lines 23-25 (pag 8).* See comment [1].

**[9]** *Lines 10-11 (pag 9).* When you explain which sea levels are used to obtain the probability distributions you mention that use sea levels with a given frequency (5 events/year or less in your case). This is equivalent to perform an extreme analysis in which you use a subset of your data composed by extreme events. Then, the usual

way should be to select sea-level events by applying the POT method using a given threshold (which will result in a varying number of events per year that, in your case, is up to five events per year) and then fitting the obtained subset by a probability function (exponential in your case).

**[10]** *Section 4.3.* You determine an attenuation factor for both coastal locations to derive local wave time-series from 15-year long offshore measurements. This is equivalent to derive an empiric wave propagation model instead of using a numerical model. However, your coastal wave time series are just 31 days long in Jätkäsaari and 11 days long in Länsikari (section 3.3). Given the short-time duration of these records, it is necessary to have more details on this analysis to trust on reconstructed long-term wave time series at both coastal sites. For instance, it should be great to have $Hs_{coastal}$-$Hs_{offshore}$ plots at both locations under different conditions $(T, \theta)$ to see the expected uncertainty in the reconstruction.

**[11]** *Line 1 (pag 11).* This is a complicated ay to say that you use the maximum run-up, $Ru_{max}$ instead of $Ru_{2\%}$.

**[12.1]** *Line 3 (pag 11).* It should be great to include a typical coastal profile of the study sites (maybe after Fig 2) to see how steep they are, especially since you are using this characteristic to approach Ru by H.

**[12.2]** The concept of *run-up height* needs to be defined to avoid misunderstandings. The run-up height is usually defined as the vertical distance between highest run-up level Ru and deepest run-down Rd. However, when we simply use wave run-up we refer to the vertical distance with respect to the mean water level. Please, clarify what you are using.

**[13]** *Line 5 (pag 11).* The use of the relationship Hmax = 2 Hs need to be justified. The ratio Hs/Hmax can be quite variable depending on local conditions (see e.g. Oliveira et al. 2018, Ocean Engineering 153, 10-22). One possibility to select the value to be used is to obtain it from the wave data recorded at your offshore location.

**[14]** The use of "full" reflection needs to be justified (or simply says that it is arbitrarily selected to be conservative). The study of Björkqvist et al (2017c) used to justify this selection was done in front of a Caisson breakwater. Since we do not know how the coast is (see comment [12.1]), it is difficult to see if the application of this reflection coefficient is appropriated for the site.

**[15]** *Section 4.5.* Since you have 15 years of simultaneous data of water level and waves, why you did not convert these series into a single series of total water level (by simple summation) and then to obtain the probability distribution. This can give you a good estimation of the "real" joint probability distribution of water levels (for all components) under current conditions. This could be used to compare with the obtained one by combining individual probability functions.

**[16]** *Section 5.* It is not clear which is the contribution of this analysis to overall results. If you are just using theoretical distributions, you do not need any data (?). However, for a real case (as it is yours) you should fit a probability distribution (Weibull in your

case) and retain the best fit (with the corresponding Weibull parameters). Of course, if you change your Weibull parameters your results will change.

You want to include here a sensitivity analysis but, there is no sensitivity analysis (nor uncertainty) associated to your previous selections (Ru formula (H), relationship between Hs and Hmax, refraction model, etc...). If you want to do a formal sensitivity analysis, probably you should account for the different contributions through the entire assessment.

**[17]** *Section 7.2. Lines 4-8 (pag 22).* See comment [15].

**[18]** *Lines 9-20 (pag 22).* This is true but this is also less and less common. As it is written, it seems that this is the most used approach. At present, flood assessments for combined water level-wave contributions, usually consider full time series instead of monthly maxima.

**[19]** *Lines 21 to 25 (pag 22).* More than the short-term variability in waves, probably, you must also consider the potential long-term variability in wave conditions for long time projections (see e.g. Méndez et al. 2006. Estimation of the long-term variability of extreme significant wave height using a time-dependent peak over threshold (pot) model." *JGR Oceans* 111,C7).

---

## Referee Comment (RC3) · Anonymous Referee #3 · 27 Feb 2018

**Review of "Combining probability distributions of sea level variations and wave run-up to evaluate coastal flooding risks"**

Manuscript # nhess-2017-438

The manuscript aims at providing estimates of extreme sea level return values based on combining probability distributions for mean sea level changes on shorter and longer time scales and wave run-up for a specific location at the Finish coast. It addresses issues relevant within the scope of the Journal and of potential interest to readers. I recommend to address the following the issues before publication.

1.  As a general comment, the approaches taken for estimating the wave run-up are rather bold and general. There are definitely a number of not necessarily better but similarly justified choices and I wonder how big the uncertainty from making such choices might be relative to issues discussed in this text. My assumption would be that it is probably a major source for uncertainty. I suggest that this should at least be discussed and conclusions should be put into perspective.

2.  I would appreciate if the authors could better motivate the sensitivity experiments described in section 5. I understand technically what was done but cannot see the added value. For the discussion of results and significance of differences, confidence intervals should be provided otherwise statements regarding the significance of the results such as on page 15, line 8 are difficult to assess.

3.  Page 5, Lines 6-7: Contribution from rivers to the water balance in particular the seasonal or longer variability should be mentioned.

4.  Page 5, Line 19: There are higher waves reported for the North Sea in chapter 7 of "State and Evolution of the Baltic Sea, 1952-2005: A Detailed 50-Year Survey of Meteorology and Climate, Physics, Chemistry, Biology, and Marine Environment" (doi: 10.1002/9780470283134)

5.  Page 7, Lines 4, 5: Please use projections instead of predictions here and at several other places in the manuscript.

6.  Page 7, Figure 3 and Lines 1-3: Please explain a bit more detailed. I cannot immediately infer the numbers given in the text from the Figure. Please also mention the baseline; that is, the year relative to which changes were computed.

7.  Figure 4: I would appreciate a comment on the extent to which the extrapolation is justified. The data seem to suggest an upper (physically based?) limit of about 150 cm.

8.  Page 12, Line 23: The authors introduce "SL-distribution" to refer to sea level variations but mainly use "still water levels" hereafter. This should be made consistent.

9.  Page 13, Table 1: Prediction should be replaced by projection. Confidence intervals would be helpful.

10. Section 8 "conclusions" is rather a summary of results.

11. Page 23, Line 14: It could also be that none of them is eventually realized.

---

## Author Comment (AC1) · 7 Jun 2018

**Referee Comments #1**

| Comment | Authors response and changes in manuscript |
|---|---|
| Major points | |
| 1. In page 9, it is written: "The exponential function was fitted to sea levels with a frequency of exceedance of 5 events/year or less." Why? The frequency of exceedance of the observed data in Figure 4 is from 1/46 to about 8000 events/year. It is thought that only the data of low frequency of exceedance are used in the curve fitting because we are interested in the events of high sea level. The reason why the data of low frequency of exceedance are used should be explained. | The exponential function was applied to the sea level distribution in order to estimate the frequencies of rare/high sea levels. The limit of 5 events/year was chosen because only the tail part of the distribution follows the exponential shape, not the entire distribution. In this way, the fit is also only done on sea level representing rare events, which may behave differently from the more frequent sea levels.

Above mentioned explanations will be added to the manuscript as follows: "We extrapolated the ccdf with an exponential function fitted to the tail of the ccdf (Fig. 4). The exponential function was fitted to sea levels with a frequency of exceedance less than $5.7 \times 10^{-4}$, which corresponds to 5 hours/year. This limit was selected because only the tail part of the distribution follows the exponential shape, while more frequent sea levels behave differently." |
| 2. In Figure 4, the maximum frequency of exceedance occurs at the sea level of -50 cm, indicating that negative storm surges frequently occur in the study area. The reason for this should be explained in the paper. | We explain in Chapter 2 (Components contributing to the sea surface level) that short-term sea level varies from -1.3 m to +2.0 m around the long-term mean sea level on the Finnish coast, and that these changes are mainly due to wind and air pressure variations. Thus -50 cm (in Figure 4) fits inside this range and is normal behaviour in the study area. |
| 3. In page 10-11, it is written: "The wave run-up can be calculated for different percentages, e.g. as the water level exceeded 2% of the time. We set out to seek a conservative estimate for the level exceeded once during the one hour time period." In the design of coastal defense structures, it is common to use the 2% run-up height to determine the crest freeboard. If the mean wave period is 8 s, the wave run-up exceeding 2% run-up height occurs 9 times during one hour, whereas the run-up height exceeded only once during one hour is exceeded 0.22% of the time. Therefore, taking the run-up height exceeded once during the one hour time period is too conservative from the engineering point of view. | We aim at estimating maximum total water level exceeded during one hour period. Thus we defined the wave run-up using the highest single wave during an hour, since this corresponds to one well defined event when the wave data and hourly water level data are combined statistically.

See also our response to the comment [11] from Reviewer #2. |
| 4. In page 11, the relationship $H_{max} = 2H_s$ is used. Longuet-Higgins (1952, J. Marine Res. 11, 246-266) presented the relationship $H_{max} = 0.707\sqrt{lnN}H_s$ for a storm with a relatively large number of waves $N$. | We used Longuet-Higgins (1952) to check our results. However, we didn't find that exact relationship in the paper, but interpolated the values of the Rayleigh distribution by using the values given in the Tables. |

| | |
|---|---|
| Again, if the mean wave period is 8 s, the number of waves during one hour is 450, which gives $H_{max}$ = 1.75$H_s$. Therefore, the relationship $H_{max} = 2H_s$ may be too conservative. | The waves at the study sites are typically short. The mean zero-upcrossing period (Tz, calculated as Tm02 from the spectral moments) is around 3 seconds (3.2 s at Länsikari and 2.8 s at Jätkäsaari). This means about 1200-1300 waves during an hour, which results in Hmax being between 1.9Hs and 2Hs. We calculated this relation for the entire time series to provide an even better overview (see Figure RC_A).

We agree that this was not presented properly in the manuscript. A more rigorous justification for choosing the relationship Hmax=2Hs will therefore be added. |
| 5. The assumption of complete wave reflection from a coastal structure (i.e. $H_{runup}= H_{max}$) may also be a too conservative assumption. This assumption, however, could be justified if we take into account the effect of wave nonlinearity in shallow water (i.e. peaked crest and flat trough), which was not considered in this study. | The water at the study sites is relatively deep when considering the short waves generated by the local fetches (around 3 s at Länsikari). At Länsikari the depth is around 10 m and at Jätkäsaari it is around 13 m. Even for the longest waves the water depth is intermediate. Shallow water nonlinearities are therefore not expected to be significant.

We want to stress that the study we cited with respect to the wave reflection was made exactly at the location of Jätkäsaari. It is therefore highly representative for this study. In Björkqvist et al. (2017) the short waves were damped by the wave damping chambers. However, the longer waves were fully reflected. Since the wave damping chambers only cover a short part of the shoreline, we have to consider conditions without the presence of them. We have no reason to believe, that all the waves wouldn't be fully reflected at a pure steep wall, since we have direct measurements of full reflection of waves that were too long for the wave damping chamber to be effective. |
| 6. Sorensen (2006, Basic Coastal Engineering, 3rd ed., Springer, p. 237) presented the relationship $R_p = R_s\sqrt{\ln(1/p)/2}$ where $R_p$ is the wave run-up height of the exceedance probability $p$ and $R_s$ is the run-up height of the incident significant wave height as if it were a monochromatic wave. If we use $p$ = 0.02 and $R_s = H_s$ (i.e. complete wave reflection), $H_{runup} = R_{2\%}$ = 1.4$H_s$ which is 70% of the value used in this study. On the other hand, if we use $p$ = 0.0022, which is the exceedance probability of the wave height exceeded only once during one hour (when the mean wave period is 8 s), $H_{runup} = R_{0.22\%}$ = 1.75$H_s$. This changes to $H_{runup} = H_{max}$ (using the relationship $H_{max}$ = | A lot of this has already been addressed, but in conclusion:

1) The choice of Hmax instead of e.g. 2% exceedance value is not a matter of being conservative. It is a choice done to get the results to correspond to "one event". It would be possible to choose a lower value that is exceeded e.g. 25 times. However, when combined with the sea level data the values would not be events, but "25 events", and the probability of 0.4% would not correspond to one event in 250 years, but to 25 events in 250 years and would inevitably lead to some inference challenges. |

| | |
|---|---|
| $0.707\sqrt{lnN}H_s$), which is the same as the run-up height used in this study except that $H_{max}$ is not calculated as $2H_s$ but as $1.75H_s$. In conclusion, to avoid too conservative estimate for wave run-up height, either $H_{runup}$ = $1.4H_s$ (general design standard) or $H_{runup}$ = $1.75H_s$ (run-up height exceeded once during one hour as taken in this study) should be used. | 2) The relation Hmax=2*Hs is not really conservative assumption. It has its bases in the measurements and theory (Rayleigh distribution). This will be clarified in the manuscript also.

3) The assumption of full reflection is the main conservative assumption. However, we feel it has a valid base, since we have observed fully reflected waves even when wave damping chambers are present. Since the damping chambers are not present everywhere, it is reasonable to assume that the short waves – that were damped by the chambers in the measurements – will be reflected in the same way as the longer waves. This might not be true, but since we have no evidence of the contrary, we feel that this is a valid assumption, albeit a conservative one. |
| 7. In addition to Table 1, it may be worthwhile to show the curves of $F_{SL}$ for 2017, 2050, and 2100. | The curve for the still water level in 2017 as well as for the years 2050 and 2100 at the Helsinki tide gauge are presented in Figure 8 in the manuscript. |
| 8. Two-parameter Weibull distributions are used for the sensitivity analysis. It may be better to add the fitted Weibull distributions (along with the shape and scale parameters) in Figure 5 to show that the Weibull distribution fits well the observation. | See our response to comment [16] from #2 Reviewer.

To provide a better comparison possibility between case study wave run-up distributions (Figure 5) and the theoretical wave run-up distributions, we plotted the theoretical wave run-up distributions also in a form of complementary cumulative distribution (see Figure RC_B) and this redrawn figure will be added to the manuscript. |
| Minor points | |
| 1. 1st line below Eq. (1): wave height >> wave run-up height | This terminological mistake will be corrected to the text where the terms of equation (1) are explained i.e. "wave height" will be changed to "wave run-up".

See also our response to comment [12.2] from #2 Reviewer. |

**Figure RC_A**. The ratio between the highest single wave and the significant wave height estimated from the Rayleigh distribution at Jätkäsaari and Länsikari.

**Figure RC_B**. Pdfs (on the left) and ccdfs (on the right) for the still water level and the six theoretical wave run-up distributions.

**Figure RC_C**. Wave run-up distributions for the two locations in the Helsinki archipelago: Jätkäsaari and Länsikari.

**Figure RC_D**. The shoreline at Jätkäsaari (from Björkqvist et al., 2017). Other parts of the shoreline are of similar shape (vertical walls), but are not equipped with wave damping chambers.

**Figure RC_E**. The yearly significant wave height at the Gulf of Finland wave buoy taken from the wave hindcast of Björkqvist et al. (2018). Trends were calculated for both the ice-free statistics and the ice-included statistics. Neither was statistically significant.

**Figure RC_F**. Probability density functions of future mean sea level at the Helsinki tide gauge for years 2050 and 2100 and the long-term mean sea level estimate of 0.19 m for year 2017. The 5[th], 50[th] and 95[th] percentiles are shown for 2050 and 2100. The data in the Figure is from the results of Pellikka et al. (2018).

---

## Author Comment (AC2) · 7 Jun 2018

**Referee Comments #2**

| Comment | Authors response and changes in manuscript |
|---------|-------------------------------------------|
| **[1]** The first comment is purely formal. Authors state in lines 22-23 (pag 3) that they are going to call "run-up level" to the combined water elevation (mean water level and wave run-up contributions. This is misleading since it is not the unusual approach in the literature. It should be better to use something like "total water level" to avoid confusion with the standard wave-induced run-up. | After re-consideration of the terms used in the manuscript we agree that using "run-up level" to represent the combination of still water level and wave run-up might be misleading and cause confusion with the wave related run-up. Thus, we will replace "run-up level" with "total water level" throughout the manuscript as suggested. |
| **[2]** *Lines 13-14 (pag 3).* Coastal floods are also a consequence of storm-surges. Please rephrase the sentence. | We agree that coastal floods are also a consequence of storm surges and that the sentence is not properly formulated. As this sentence is not very relevant for our introduction (which is already quite long), we decided that the whole sentence will be removed from the manuscript. |
| **[3]** *Lines 6 (pag 4).* In general terms, the wave-induced component of the water level at the shoreline is the run-up and not the wave height (a different thing is that you approach the run-up with the wave height but this depends on how you calculate it). | This terminological mistake will be corrected to the text where the terms of equation (1) are explained i.e. "wave height" will be changed to "wave run-up".

See also our response to your comment [12.2]. |
| **[4]** *Line 1 (pag 5).* Long-term mean sea level does not change from decade to decade. Mean sea level is continuously varying and "long-term" refers to the low-frequency component which, apparently, you consider to be associated to periods in the scale of decades (or longer). | We agree that the sentence was poorly formulated. However our purpose in the manuscript is to distinguish between the sea level variations taking place at short time scale (e.g. storm surges) and those that happen slowly within long time span (e.g. mean sea level change).

The sentence will be reformulated in a following manner: "The long-term mean sea level on the Finnish coast, on decadal time scale, is affected by the global mean sea level, the post-glacial land uplift and the Baltic Sea water balance (Johansson et al., 2014)." |
| **[5]** *Section 3.1. Long-term sea level.* You are using long-term estimations of sea level at selected horizons based on a paper that is under review. If this component is important for your calculations, it can be difficult for some readers to trust on it without having access to the scientific work supporting used values. | The paper we are referring to has now been published. The reference list will be updated accordingly:

Pellikka, H., Leijala, U., Johansson, M. M., Leinonen, K., Kahma, K. K., 2018. Future probabilities of coastal floods in Finland. Continental Shelf Research, 157, 32-42. DOI: 10.1016/j.csr.2018.02.006. |
| **[6]** *Section 3.2*. Please change the heading. Here you are not describing variability but just the existing data. They are simply water level measurements acquired by using tidal gauges. Use something similar | We agree that the heading was too complicated as the aim of this section is to just describe the tide gauge data. Thus we will change the heading of Section 3.2 simply to "Sea level data". |

| | |
|---|---|
| to heading of section 3.3 (e.g. tidal data, water level data). | |
| **[7]** *Lines 6-7 (Page 8).* Please remove the last sentence "The significant wave height is …". If you want to use a definition of Hs use a formal one (e.g. based on spectral moments). | We will replace the formal definition given on page 8, line 4 with the one using spectral moments. The "layman" definition will also be removed as redundant, as you suggested. |
| **[8]** *Lines 23-25 (pag 8)*. See comment [1]. | See our response to your comment [1]. |
| **[9]** *Lines 10-11 (pag 9).* When you explain which sea levels are used to obtain the probability distributions you mention that use sea levels with a given frequency (5 events/year or less in your case). This is equivalent to perform an extreme analysis in which you use a subset of your data composed by extreme events. Then, the usual way should be to select sea-level events by applying the POT method using a given threshold (which will result in a varying number of events per year that, in your case, is up to five events per year) and then fitting the obtained subset by a probability function (exponential in your case). | In the POT method two limits are usually set. One is the threshold (e.g. a certain sea level value), which will give us a certain amount of events per year (on average). The second limit determines the distance between two points. This second limit is set to remove events that are not independent, which enables the final data set to converge to a Pareto distribution. The second limit can be in the order of 24-72 hours, but for sea level data in the Baltic Sea the correlation might be significantly longer (in the order of months). This is because the slow changes in the total water volume in the Baltic Sea.

If a proper POT method is applied, the resulting distribution converges to a generalized Pareto distribution (GPD) and will no longer simply be the tail of the original distribution. The main point is the following: in order to use our method, we ultimately need to revert back to the full distribution, since the statistical combination with the wave run-up will otherwise not be possible. If we have fitted a GPD to the data we got by applying the POT method and use that tail to extrapolate the original data, then we are extrapolating the original distribution with a fit that has been made to a different distribution (the GPD). This is obviously something we want to avoid.

By fitting the exponential distribution to the tail, we are essentially using the POT method to the extent it is possible in our case. Using the POT method "to its fullest" would change the distribution, since the entire point with the method is to converge the subset of the original data to a GPD. Since we are not only interested in the extreme values, but need the full distribution to combine the sea level data with the wave run-up, the traditional use of the POT method is not a suitable tool for our purposes.

See also our response to comment [1] from #1 Reviewer. |

| | |
|---|---|
| **[10]** *Section 4.3*. You determine an attenuation factor for both coastal locations to derive local wave time-series from 15-year long offshore measurements. This is equivalent to derive an empiric wave propagation model instead of using a numerical model. However, your coastal wave time series are just 31 days long in Jätkäsaari and 11 days long in Länsikari (section 3.3). Given the short-time duration of these records, it is necessary to have more details on this analysis to trust on reconstructed long-term wave time series at both coastal sites. For instance, it should be great to have Hs coastal-Hs offshore plots at both locations under different conditions (T, θ) to see the expected uncertainty in the reconstruction. | First, we want to stress that the attenuated time series we get with the transfer function is not a valid realisation of the wave height time series for the entire 15 year period. The main idea is, that while we can get the typical values (although with a slight positive bias because the measurements were made in the autumn) directly from the measurements, we cannot get the rare exceedances. We therefore determined a transfer function that was adjusted to accurately model the highest values in the measurement time series. When used on the longer open sea wave buoy measurements we can then get information about the nearshore wave height during the more extreme wave events that have happened outside of our short measurement period.

Figure RC_C shows that the estimated distribution from the transfer function coincides with the observed values at the tail of the observed distribution.

We acknowledge that this is not an optimal way, but it was a practical solution to extract as much information from the existing data set as possible. However, the method presented in the paper is in no way reliant on the method we used to determine the wave height distribution.

We will add the information shown in Figure RC_C also to the manuscript, in order to show in more detail how the wave run-up distributions were formed. |
| **[11]** *Line 1 (pag 11).* This is a complicated ay to say that you use the maximum run-up, $Ru_{max}$ instead of $Ru_{2\%}$. | Our purpose is to say that we use the maximum run-up and we acknowledge that our way of saying this could be more straightforward. We will rephrase the explanation as follows: "The final step was to estimate the wave run-up, i.e. the maximum vertical elevation of the water in relation to the still water level. We defined the wave run-up using the highest single wave during an hour, since this will produce one well defined event when combined statistically with the water level data." This sentence is followed by more detailed explanation of the selected method. |
| **[12.1]** *Line 3 (pag 11).* It should be great to include a typical coastal profile of the study sites (maybe after Fig 2) to see how steep they are, especially since you are using this characteristic to approach Ru by H. | We have added a picture of the shoreline at Jätkäsaari (Figure RC_D, from Björkqvist et al., 2017). Although this figure shows the wave damping chambers, the shoreline is similar at other locations |

| | that are not equipped with wave damping chambers (see Figure 1 in Björkqvist et al., 2017). |
|---|---|
| **[12.2]** The concept of *run-up height* needs to be defined to avoid misunderstandings. The run-up height is usually defined as the vertical distance between highest run-up level Ru and deepest run-down Rd. However, when we simply use wave run-up we refer to the vertical distance with respect to the mean water level. Please, clarify what you are using. | We agree that concepts need to be clearly and uniformly defined throughout the paper. In this study we define the "run-up" as the maximum vertical elevation of the water in relation to the still water level during a certain period. We will define this clearly in the manuscript and remove "run-up height" definition to avoid misunderstandings. |
| **[13]** *Line 5 (pag 11).* The use of the relationship Hmax = 2 Hs need to be justified. The ratio Hs/Hmax can be quite variable depending on local conditions (see e.g. Oliveira et al. 2018, Ocean Engineering 153, 10-22). One possibility to select the value to be used is to obtain it from the wave data recorded at your offshore location. | We calculated this based on the wave data recorded at the nearshore location (not the offshore location, since the typical wave periods are much longer there). The mean zero-upcrossing period (Tz, calculated as Tm02 from the spectral moments) are around 3 seconds (3.2 s at Länsikari and 2.8 s at Jätkäsaari). This means about 1200-1300 waves during an hour, which results in Hmax being between 1.9Hs and 2Hs (Figure RC_A).

See also our response to the fourth comment from Reviewer #1. |
| **[14]** The use of "full" reflection needs to be justified (or simply says that it is arbitrarily selected to be conservative). The study of Björkqvist et al (2017c) used to justify this selection was done in front of a Caisson breakwater. Since we do not know how the coast is (see comment [12.1]), it is difficult to see if the application of this reflection coefficient is appropriated for the site. | See our response to your comment [12.1]. |
| **[15]** *Section 4.5.* Since you have 15 years of simultaneous data of water level and waves, why you did not convert these series into a single series of total water level (by simple summation) and then to obtain the probability distribution. This can give you a good estimation of the "real" joint probability distribution of water levels (for all components) under current conditions. This could be used to compare with the obtained one by combining individual probability functions. | This is an excellent point, which we would certainly have done if we had the data to do it. However, as addressed in point [10], the wave heights that are estimated do not produce a proper time series, but are used to complete the tail of the distribution based on the measurements (see Figure RC_C). Since the transfer function is constructed with the aim to get the highest tail (not to e.g. minimize the bias), it means that the lower wave heights are overestimated. This is acceptable, since they are not used to construct the distribution. |
| **[16]** *Section 5.* It is not clear which is the contribution of this analysis to overall results. If you are just using theoretical distributions, you do not need any data (?). However, for a real case (as it is yours) you should fit a probability distribution (Weibull in your | This is a relevant point. The purpose of the "sensitivity test" is to study how different wave height conditions (based on theoretical wave run-up distributions) affect the total water level when the still water level distribution is kept unchanged.

We agree that the contribution of this section to the overall results is not clear, and calling it a "sensitivity |

| | |
|---|---|
| case) and retain the best fit (with the corresponding Weibull parameters). Of course, if you change your Weibull parameters your results will change.

You want to include here a sensitivity analysis but, there is no sensitivity analysis (nor uncertainty) associated to your previous selections (Ru formula (H), relationship between Hs and Hmax, refraction model, etc…). If you want to do a formal sensitivity analysis, probably you should account for the different contributions through the entire assessment. | test" is misleading. In order to clarify our aim of the analysis we will remove Section 5 and instead reorganize the "Results" section to include: 1) the case study at Helsinki, 2) the study with these theoretical wave distributions, and 3) a comparison between the results of these. |
| **[17]** *Section 7.2. Lines 4-8 (pag 22).* See comment [15]. | See our response to your comment [15]. |
| **[18]** *Lines 9-20 (pag 22).* This is true but this is also less and less common. As it is written, it seems that this is the most used approach. At present, flood assessments for combined water level-wave contributions, usually consider full time series instead of monthly maxima. | We acknowledge that we generalized unnecessarily the use of block maxima. We will rephrase the sentence: "Using block maxima of sea level variations — such as the monthly maxima used by Pellikka et al. (2018) — in our analysis would implicitly restrict the study of the joint effect to cases where the still water level is high, thus excluding combinations of moderate still water level and high waves." |
| **[19]** *Lines 21 to 25 (pag 22).* More than the short-term variability in waves, probably, you must also consider the potential long-term variability in wave conditions for long time projections (see e.g. Méndez et al. 2006. Estimation of the long-term variability of extreme significant wave height using a time-dependent peak over threshold (pot) model." *JGR Oceans* 111,C7). | Using the verified wave model data from Björkqvist et al. (2018) we calculated the mean significant wave height at the GoF wave buoy for the years 1965-2005 (the hindcast cannot resolve the nearshore conditions of Länsikari and Jätkäsaari).

The results are shown in Figure RC_E for both ice-free statistics and ice-included (as Hs=0) statistics. In both statistics the trend is small, and not statistically significant according to a t-test.

This is supported by Kudryavtseva and Soomere (2017). The authors used satellite altimetry data (1996-2015) and found no statistically significant trend in the Gulf of Finland.

Of course, the absence of evidence is not evidence of absence, but using the current knowledge we have no means to predict the future changes of the significant wave height in the GoF. We have mentioned using long-term scenarios for wave conditions in the Discussion part of the manuscript, as a potential improvement on our method in future studies. |

| | Méndez et al. (2006) used a POT-method where the coefficients of the GDP-distribution where allowed to vary in time. The time varying parameters can capture some of the seasonal variability that is lost if the POT-method is used on the entire data set with only one set of parameters. However, since the method implemented in this paper uses the full distributions, all seasonal variations are already present in the data, and no special methods are required to account for them. |
|---|---|

[Figure]

**Figure RC_A**. The ratio between the highest single wave and the significant wave height estimated from the Rayleigh distribution at Jätkäsaari and Länsikari.

[Figure]

**Figure RC_B**. Pdfs (on the left) and ccdfs (on the right) for the still water level and the six theoretical wave run-up distributions.

[Figure]

**Figure RC_C**. Wave run-up distributions for the two locations in the Helsinki archipelago: Jätkäsaari and Länsikari.

[Figure]

**Figure RC_D**. The shoreline at Jätkäsaari (from Björkqvist et al., 2017). Other parts of the shoreline are of similar shape (vertical walls), but are not equipped with wave damping chambers.

[Figure]

**Figure RC_E**. The yearly significant wave height at the Gulf of Finland wave buoy taken from the wave hindcast of Björkqvist et al. (2018). Trends were calculated for both the ice-free statistics and the ice-included statistics. Neither was statistically significant.

[Figure]

**Figure RC_F**. Probability density functions of future mean sea level at the Helsinki tide gauge for years 2050 and 2100 and the long-term mean sea level estimate of 0.19 m for year 2017. The 5[th], 50[th] and 95[th] percentiles are shown for 2050 and 2100. The data in the Figure is from the results of Pellikka et al. (2018).

---

## Author Comment (AC3) · 7 Jun 2018

**Referee Comments #3**

| Comment | Authors response and changes in manuscript |
|---|---|
| 1. As a general comment, the approaches taken for estimating the wave run-up are rather bold and general. There are definitely a number of not necessarily better but similarly justified choices and I wonder how big the uncertainty from making such choices might be relative to issues discussed in this text. My assumption would be that it is probably a major source for uncertainty. I suggest that this should at least be discussed and conclusions should be put into perspective. | You are correct that we have not justified our choices properly. Taking into account also the feedback from reviewers #2 and #1, there are three main points we need to address:

1) Using the maximum wave as the run-up instead of a value exceeded e.g. 2% of the time.
2) The Hmax=2*Hs approximation.
3) The assumption of full reflection.

We will repeat our response to Reviewer #1 below:

1) The choice of Hmax instead of e.g. 2% exceedance value is not a matter of being conservative. It is a choice done to get the results to correspond to "one event". It would be possible to choose a lower value that is exceeded e.g. 25 times. However, when combined with the sea level data the values would not be events, but "25 events", and the probability of 0.4% would not correspond to one event in 250 years, but to 25 events in 250 years and would inevitably lead to some inference challenges.

2) The relation Hmax=2*Hs is not really a conservative assumption. It has its bases in the measurements and theory (Rayleigh distribution). This will be clarified in the manuscript also.

3) The assumption of full reflection is the main conservative assumption. However, we feel it has a valid base, since we have observed fully reflected waves even when wave damping chambers are present. Since the damping chambers are not present everywhere, it is reasonable to assume that the short waves – that were damped by the chambers in the measurements – will be reflected in the same way as the longer waves. This might not be true, but since we have no evidence of the contrary, we feel that this is a valid assumption, albeit a conservative one.

We will modify the manuscript to better explain the reasons for 1), better justify the validity of 2) and discuss the assumption taken to conclude 3). |

| | |
|---|---|
| 2. I would appreciate if the authors could better motivate the sensitivity experiments described in section 5. I understand technically what was done but cannot see the added value. For the discussion of results and significance of differences, confidence intervals should be provided otherwise statements regarding the significance of the results such as on page 15, line 8 are difficult to assess. | See our response to comment [16] from #2 Reviewer.

Our purpose is not to refer to statistical significance in the discussion of results and significance of differences, and we will modify our statements according to this in the "Results" section by e.g. replacing the term "significant" with more appropriate expression (e.g. "The contribution of the waves is now *larger compared to* the situation with the first pair.") |
| 3. Page 5, Lines 6-7: Contribution from rivers to the water balance in particular the seasonal or longer variability should be mentioned. | We will add a mentioning of the contribution of rivers to the water balance in Chapter 2 (in the section describing the long-term mean sea level). |
| 4. Page 5, Line 19: There are higher waves reported for the North Sea in chapter 7 of "State and Evolution of the Baltic Sea, 1952-2005: A Detailed 50-Year Survey of Meteorology and Climate, Physics, Chemistry, Biology, and Marine Environment" (doi: 10.1002/9780470283134) | It is true that there are higher waves outside the Baltic Sea, which is a semi-enclosed basin and rather shallow compared to other seas around the globe. At this point we did not have an access to the referred book but as this paper focuses on the Baltic Sea we therefore in the text have referred to the highest wave that has been measured inside the Baltic Sea (see Björkqvist et al., 2017b). |
| 5. Page 7, Lines 4, 5: Please use projections instead of predictions here and at several other places in the manuscript. | We agree that projections is better term for the mean sea level scenarios and this will be corrected throughout the manuscript as suggested. |
| 6. Page 7, Figure 3 and Lines 1-3: Please explain a bit more detailed. I cannot immediately infer the numbers given in the text from the Figure. Please also mention the baseline; that is, the year relative to which changes were computed. | We agree that our explanation could be more detailed and clear.

The purpose of Figure 3 is to show the different shapes of the mean sea level probability density functions for the selected years i.e. the spreading of the distribution towards the future. Figure 3 is drawn from the results of Pellikka et al. (2018) that is published now:

Pellikka, H., Leijala, U., Johansson, M. M., Leinonen, K., Kahma, K. K., 2018. Future probabilities of coastal floods in Finland. Continental Shelf Research, 157, 32-42. DOI: 10.1016/j.csr.2018.02.006.

To clarify our message, we have redrawn Figure 3 and rephrased its caption (see Figure RC_F) so that the numbers mentioned in the text can be read from the figure. The manuscript text will be changed accordingly, and the baseline will be also mentioned. |
| 7. Figure 4: I would appreciate a comment on the extent to which the extrapolation is justified. The | We don't know of any physical upper limit for the short-term sea level variations. The value that seems to be the upper limit in Figure 4 (around 150 cm) is |

| | |
|---|---|
| data seem to suggest an upper (physically based?) limit of about 150 cm. | due to the fact that the highest observed points are not independent but originate from the same sea level event which lasted for several hours.

We have addressed our decision to use the exponential fit by referring to studies of Särkkä et al., 2017 in the text.

See also our response to the comment [1] from #1 Reviewer. |
| 8. Page 12, Line 23: The authors introduce "SL-distribution" to refer to sea level variations but mainly use "still water levels" hereafter. This should be made consistent. | This is a good comment. We agree that "SL-distribution" is unnecessary definition and using it complicates the text. Thus we will rephrase the sentences that involve SL-distribution and used $F_{SL}$ instead. The same procedure will be done for sentences including "SL,W-distribution" (i.e. SL,W-distribution will be replaced by $F_{SL,W}$). |
| 9. Page 13, Table 1: Prediction should be replaced by projection. Confidence intervals would be helpful. | We will replace "prediction" with "projection" as suggested (see also our response to your comment [5]).

We agree that confidence intervals would be helpful. However, calculating them would require more in depth analysis of the uncertainties of the sea level distributions (short and long-term), which we decided to leave outside this study where the main focus is to present the method for combining the sea level distributions with the wave distributions. |
| 10. Section 8 "conclusions" is rather a summary of results. | We agree that the "Conclusions" section was mainly summarizing our results. Thus we will rewrite it to better address conclusions that arise from our results. |
| 11. Page 23, Line 14: It could also be that none of them is eventually realized. | This is a good point and true. The sentence will be reformulated better. |

**Figure RC_A**. The ratio between the highest single wave and the significant wave height estimated from the Rayleigh distribution at Jätkäsaari and Länsikari.

**Figure RC_B**. Pdfs (on the left) and ccdfs (on the right) for the still water level and the six theoretical wave run-up distributions.

**Figure RC_C**. Wave run-up distributions for the two locations in the Helsinki archipelago: Jätkäsaari and Länsikari.

**Figure RC_D**. The shoreline at Jätkäsaari (from Björkqvist et al., 2017). Other parts of the shoreline are of similar shape (vertical walls), but are not equipped with wave damping chambers.

**Figure RC_E**. The yearly significant wave height at the Gulf of Finland wave buoy taken from the wave hindcast of Björkqvist et al. (2018). Trends were calculated for both the ice-free statistics and the ice-included statistics. Neither was statistically significant.

**Figure RC_F**. Probability density functions of future mean sea level at the Helsinki tide gauge for years 2050 and 2100 and the long-term mean sea level estimate of 0.19 m for year 2017. The 5th, 50th and 95th percentiles are shown for 2050 and 2100. The data in the Figure is from the results of Pellikka et al. (2018).